# TokenTune: Dual-Level Utility Estimation for Scalable Data Selection in Instruction Tuning

## Abstract

Recent studies indicate that data quality is more important than quantity for fine-tuning of large language models (LLMs). However, existing data selection methods face two key limitations. First, they lack an effective utility estimation function: *sample-level* utility computes the score for entire examples but ignores which tokens are actually useful, while *token-level* methods drop tokens with multiple valid answers and thus remove valuable learning signals. Second, these methods are inefficient because they require full-dataset inference to compute utilities, making them prohibitively expensive at scale. To address these challenges, we propose TokenTune, an efficient data selection framework for instruction tuning. The key idea of TokenTune is a dual-level utility function that operates at both the token and sample levels. At the token level, it identifies learnable tokens that still provide strong gradient signals and multi-answer tokens that preserve diversity under incomplete supervision. At the sample level, it derives a utility score directly from token signals, *avoiding redundant full-dataset inference*. To further scale, TokenTune employs a two-stage design. In the selection stage, a multi-armed bandit adaptively prioritizes informative clusters, from which high-utility samples are chosen using the sample-level score. In the training stage, the token-level utility guides gated optimization: learnable tokens strengthen supervision, while multi-answer tokens preserve diversity. Extensive experiments across 7 benchmarks show that TokenTune significantly outperforms state-of-the-art methods, improving average performance by +3.8% while using only 5% of the full training data and reducing overall training time by 8-10×.

## 1 Introduction

Instruction tuning has emerged as a powerful paradigm to improve the performance and alignment of large language models (LLMs) by fine-tuning them on instruction–response pairs (Sun et al., 2024; Li et al., 2024b; Chang et al., 2024). Recent studies show that data quality, rather than sheer quantity, is crucial for substantial performance gains (Zhou et al., 2023; Albalak et al.). This insight has motivated a growing line of work on **data selection methods**, which aim to identify and prioritize informative subsets of training data automatically. So far, most existing approaches have operated *at the sample level* (Li et al., 2024a; Han et al.; Lin et al., 2025), where each instruction–response pair is treated as a single unit with one utility score. However, this **sample-level-only selection** evaluates each example as a whole and ignores variation among its tokens. In practice, a sample with a high overall score may still contain many uninformative or noisy tokens, which *weaken gradient signals and waste the limited training budget*. To mitigate this limitation, recent work has shifted toward **token-level-only selection** (Pang et al., 2025), aiming to remove noisy tokens within samples. Token-level filtering cleans up noisy tokens but does not capture whether the remaining content still forms a meaningful and instructive example. As a result, it may mistakenly discard globally valuable samples whose usefulness emerges only when the full context is preserved.

This predicament leads to our research question: *Can we design a unified framework that unifies the strengths of token-level granularity and sample-level context to better select high-quality data?*

**Intuitive Dual-Level Selection Approaches.** As shown in Figure 1, there are two intuitive methods to achieve this goal. Sample-to-Token (S2T) first selects samples and then cleans tokens within the chosen subset, whereas Token-to-Sample (T2S) reverses this order by first scoring tokens and then

Figure 1: Comparison of different data selection paradigms. (a) Sample-to-Token (S2T) and (b) Token-to-Sample (T2S) represent straightforward combinations of sample- and token-level selection. Both require two-stage estimation, leading to higher latency and suboptimal performance. (c) Our TokenTune achieves dual-level selection without redundant inference by composing sample utilities directly from token signals and leveraging clustering-based scheduling.

aggregating them to estimate sample utility. However, although these strategies can achieve *effective* selection, they still expose several unresolved **challenges** in terms of *efficiency* and *generalization*.

**Challenges. First**, while dual-level scoring improves selection quality, it incurs significant computational overhead since it requires computing utility at both the sample and token levels. This effectively doubles the computational cost per example (**C1**). **Second**, even if redundant scoring is avoided, computing utility for all samples in a large-scale dataset is still prohibitively expensive. We term this issue as *scalability* challenges (**C2**). If we can focus scoring only on a small set of promising samples, we can significantly accelerate the overall selection process without compromising performance. **Third**, once a high-quality subset with valuable tokens is selected, the standard supervised fine-tuning (SFT) paradigm typically applies cross-entropy loss uniformly across all selected tokens. However, this loss function assumes that each token position has only one correct answer, represented by a one-hot target distribution. In instruction tuning, many output positions admit multiple plausible candidate tokens. Treating such multi-answer tokens as only one correct answer can penalize valid alternatives, causing the model to overfit to single references and reducing its ability to generate diverse outputs. This challenge comprises two key aspects: (1) how to *identify multi-answer tokens* during selection (**C3.1**), and (2) how to *optimize these multi-answer tokens* with a more flexible supervision strategy than hard one-hot cross-entropy (**C3.2**).

**Our Methodology.** To tackle the above challenges, we propose TokenTune, a dual-level framework for efficient and generalizable data selection. At its core, TokenTune proposes an effective dual-level utility function operates at both the token and sample level. At the token level, it introduces two complementary indicators to identify learnable tokens and multi-answer tokens (addressing **C3.1**), At the sample level, it derives a sample utility score directly from these token-level signals, avoiding redundant sample scoring process (addressing **C1**). To ensure scalability (C2) and generalization (C3), TokenTune employs a two-stage pipeline built upon this dual-level utility function. In the selection stage, TokenTune integrates a multi-armed bandit scheduler that adaptively prioritizes informative clusters, focusing utility scoring process on high-utility subsets of the data pool (addressing **C2**). In the training stage, a gated optimization strategy is proposed to route tokens into distinct optimization objectives. Specifically, TokenTune uses cross-entropy loss for learnable tokens to provide strong and reliable supervision that guides downstream task learning, while utilizing self-distillation on multi-answer tokens to maintain output diversity (addressing **C3.2**).

**Contributions.** This paper makes the following contributions:

- We propose TokenTune, which combines a multi-armed bandit scheduler for scalable and informative sample selection with a token-aware gated optimization strategy that routes different token types to distinct training objectives. (Section 2)

- We design a dual-level utility function that jointly operates at the token and sample levels. It leverages token-level indicators (*learning gain* for learnable tokens and *answer uncertainty*

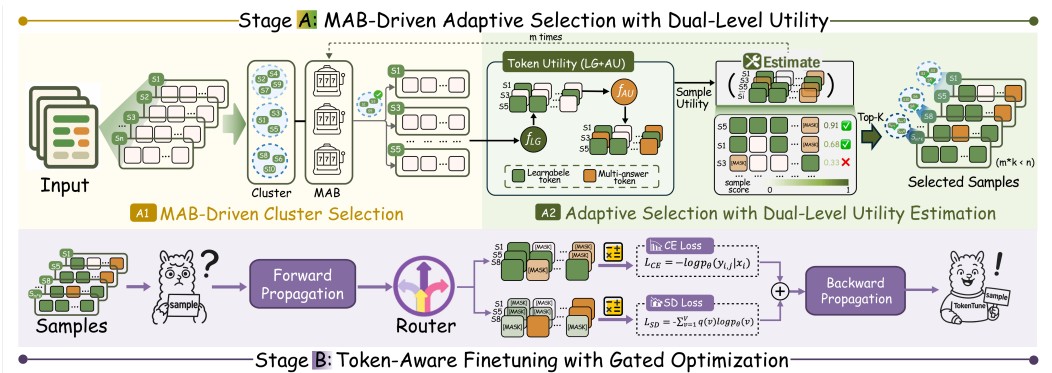

Figure 2: An Overview of TOKENTUNE.

for multi-answer tokens) and derives sample-level utilities directly from these signals, avoiding redundant inference. (Section 2.2)

- We provide theoretical insights of TOKENTUNE, showing why our token-level indicators capture learnable and uncertain tokens, why their aggregation yields effective sample-level selection, and why uncertainty-aware objectives improve training robustness. (Section 3)

- Extensive experiments across diverse benchmarks show that TOKENTUNE significantly outperforms state-of-the-art methods, improving average model performance by approximately 3.8% while using only 5% of the training data and reducing overall training time by 8-10×. (Section 4)

## 2 TOKENTUNE

### 2.1 TOKENTUNE OVERVIEW

**Core Components of TOKENTUNE.** TOKENTUNE has three core components: ❶ a *dual-level utility function* that jointly captures token- and sample-level utilities by leveraging token indicators and deriving sample-level scores without redundant inference; ❷ an *adaptive MAB scheduler* that partitions the data pool into clusters and uses a multi-armed bandit to prioritize promising regions for scalable selection; and ❸ a *token-aware gated optimization* strategy that differentiates token roles during training, assigning cross-entropy to learnable tokens, self-distillation to multi-answer tokens, and suppressing uninformative tokens.

As shown in Figure 2, TOKENTUNE follows a two-stage process. **Stage 1** adaptively selects high-utility samples by first clustering the pool, then applying the bandit to focus on promising clusters, and finally conducting dual-level utility estimation within them. **Stage 2** finetunes on the selected subset with token-aware gated optimization, ensuring that retained signals—both learnable and multi-answer—contribute effectively to training.

### 2.2 DUAL-LEVEL UTILITY FUNCTION

TOKENTUNE introduces a dual-level utility function that operates at both the token and sample levels, enabling fine-grained token filtering while simultaneously guiding principled sample selection.

#### 2.2.1 TOKEN-LEVEL UTILITY FUNCTION

Our token-level utility function are designed to answer a simple question: *Given a limited training budget, on which tokens does an additional update yield the largest marginal improvement?*

To this end, we follow two principles: (i) we would like to prioritize tokens whose further training is expected to produce a large reduction in loss *per token*, so that each gradient update is

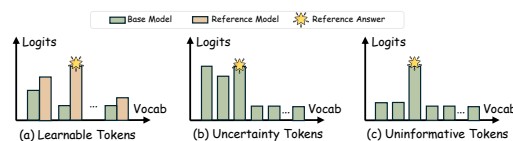

Figure 3: Different Types of Tokens.

spent where it is most effective. (ii) Among such tokens, we want to distinguish between those that are *under-learned but consistent* (single correct answer) and those that are *inherently multi-answer* (several plausible outputs), since the latter should not be forced into a single hard label.

To accomplish these principles, it involves two key components: a utility function that quantifies token informativeness, and principled decision boundaries that distinguish among the categories.

**Utility Function**: **Measuring What Matters in Tokens**. To quantify token informativeness, we define a utility function with two complementary components, namely *Learning Gain (LG)* and *Answer Uncertainty (AU)*.

**Definition 2.1** (❶ *Learning Gain (LG)*). LG quantifies how much a token stands to benefit from further training. Formally, given sample $x_i$ with $n_i$ tokens $\{x_{i,1}, \ldots, x_{i,n_i}\}$, $y_{i,j}$ the target token at position $(i,j)$. For a model with parameters $\theta$, we define its token-level loss at $(i,j)$ as

$$\ell_\theta(x_{i,j}) := -\log p_\theta(y_{i,j} \mid x_i) \tag{1}$$

Given a reference model $\theta_{ref}$ and the current model $\theta_0$, the *learning gain* at token $(i,j)$ is defined as Learning Gain $LG(x_{i,j})$. $LG$ measures whether a token remains learnable:

$$LG(x_{i,j}) := \Delta\ell(x_{i,j}) = \ell_0(x_{i,j}) - \ell_{\text{ref}}(x_{i,j}) \tag{2}$$

where $\ell_{\text{ref}}$ and $\ell_0$ denote token-level losses under the reference and current model. A large $LG(x_{i,j})$ indicates that the token remains difficult and thus provides meaningful gradient signal, whereas a small value suggests it has already been mastered by model.

**Definition 2.2** (❷ *Answer Uncertainty (AU)*). Some tokens are inherently ambiguous, admitting multiple plausible answers. To identify such cases, we model predictive uncertainty with an evidential Dirichlet distribution. For a token position $(i,j)$ with target token $x_{i,j}$, let $\mathbf{z}(x_{i,j}) = (z_1(x_{i,j}), \ldots, z_K(x_{i,j}))$ denote the pre-softmax logits over the vocabulary produced by the model. We first map logits to non-negative evidence and obtain Dirichlet parameters $\boldsymbol{\alpha}(x_{i,j}) = (\alpha_1(x_{i,j}), \ldots, \alpha_K(x_{i,j}))$. We then define the answer uncertainty at $(i,j)$ as the *expected predictive entropy* of the categorical distribution $\mathbf{p}$ drawn from this Dirichlet:

$$AU(x_{i,j}) := \mathbb{E}_{\mathbf{p} \sim \text{Dir}(\boldsymbol{\alpha}(x_{i,j})+\mathbf{1})}\Big[ -\sum_{k=1}^{K} p_k \log p_k \Big], \tag{3}$$

where $\mathbf{1}$ is the all-ones vector. Using standard properties of the Dirichlet distribution, this expectation admits the following closed-form expression in terms of the digamma function:

$$\alpha_k(x_{i,j}) = \max(0, z_k(x_{i,j})) + 1, \quad \alpha_0(x_{i,j}) = \sum_{k=1}^{K} \alpha_k(x_{i,j}),$$

$$AU(x_{i,j}) = -\sum_{k=1}^{K} \frac{\alpha_k(x_{i,j})}{\alpha_0(x_{i,j})} \Big( \psi(\alpha_k(x_{i,j}) + 1) - \psi(\alpha_0(x_{i,j}) + 1) \Big), \tag{4}$$

where $\psi(\cdot)$ is the digamma function. A high value of $AU(x_{i,j})$ indicates strong evidence for multiple plausible outputs at position $(i,j)$, suggesting that such tokens should not be optimized with hard labels. In Appendix A.1 and Appendix J.2, we have explain why AU captures inherently multi-answer tokens. Please refer to this part for more details.

**Decision Boundaries: From Token Scores to Token Labels.**

$$\hat{y}_{i,j} = \begin{cases} 1, & \text{if } LG(x_{i,j}) > \tau_{LG}, \\ 2, & \text{if } LG(x_{i,j}) \le \tau_{LG} \text{ and } AU(x_{i,j}) > \tau_{AU}, \\ 0, & \text{otherwise}, \end{cases} \tag{5}$$

where $\tau_{LG}$ and $\tau_{AU}$ are predefined thresholds. Here, $\hat{y}_{i,j} = 1$ indicates that the token will be routed to a strong supervision objective, $\hat{y}_{i,j} = 2$ indicates routing to a distillation-based objective, and $\hat{y}_{i,j} = 0$ corresponds to suppression during training.

### 2.2.2 SAMPLE-LEVEL UTILITY FUNCTION

A key challenge in model-aware data selection is how to aggregate fine-grained token-level utilities into a reliable sample-level score. Many existing methods repeatedly run the model over the entire dataset to estimate per-sample gains, which becomes computationally prohibitive at scale.

Our goal is to construct a sample-level utility that (i) reflects how much training on a sample is expected to improve the model *per token*, so that scores are comparable across samples of different

lengths, (ii) is directly induced from the token-level learning gains $LG(x_{i,j})$ rather than introducing a separate heuristic at the sample level, and (iii) requires *no additional* forward passes beyond those already used to compute token losses.

Formally, recall that for sample $x_i = \{x_{i,1}, \ldots, x_{i,n_i}\}$ and target tokens $y_{i,j}$, we defined in Eq. 2 the token-level learning gain

$$LG(x_{i,j}) := \Delta\ell(x_{i,j}) = \ell_{\text{ref}}(x_{i,j}) - \ell_0(x_{i,j}),$$

where $\ell_{\text{ref}}$ and $\ell_0$ are the token losses under the reference and current model, respectively. For notational convenience, we write

$$\Delta\ell_{i,j} := \Delta\ell(x_{i,j}), \qquad b_{i,j} := \ell_0(x_{i,j}) > 0.$$

We first define a *per-token utility density*

$$\rho_{i,j} := \frac{\Delta\ell_{i,j}}{b_{i,j}}, \tag{6}$$

which normalizes the learning gain at $(i, j)$ by its baseline loss. Here $b_{i,j}$ is simply the token-level loss under the current model. Under cross-entropy, this loss is $-\log p_{\theta_0}(y_{i,j} \mid x_i)$: tokens with large $b_{i,j}$ are those to which the model assigns low probability (i.e., it is still uncertain or often wrong), while tokens with small $b_{i,j}$ are already well mastered. We therefore interpret $b_{i,j}$ as the current *difficulty* of token $(i, j)$. The density $\rho_{i,j}$ then normalizes the learning gain at $(i, j)$ by its baseline loss. Intuitively, $\rho_{i,j}$ measures how much additional loss reduction we obtain relative to the current difficulty of this token, i.e., the loss improvement per unit difficulty. For any subset of positions $S \subseteq \{1, \ldots, n_i\}$, the *sample-level utility function* is defined as the weighted average of the token densities:

$$U_i(S) := \frac{\sum_{j \in S} \Delta\ell_{i,j}}{\sum_{j \in S} b_{i,j}} = \sum_{j \in S} w_{i,j}(S)\, \rho_{i,j}, \qquad w_{i,j}(S) := \frac{b_{i,j}}{\sum_{t \in S} b_{i,t}}, \tag{7}$$

where $w_{i,j}(S)$ is a normalized weight over tokens in $S$. In practice, we choose $S$ as the set of top-$k\%$ tokens in $x_i$ ranked by $\rho_{i,j}$, and use the resulting $U_i(S)$ as the sample-level score $U_k^{\text{Sample}}(x_i)$ for data selection. This construction reuses the token-level LG signals and does not require any extra inference beyond the losses already computed during training. We provide a more detailed derivation and discussion in Appendix J.3.

## 2.3 Adaptive Data Selection via MAB-Integrated Scheduler

To efficiently scale TokenTune to large datasets, we integrate the Multi-Armed Bandit (MAB) algorithm with our dual-level utility estimation. This scheduler adaptively explores clusters while exploiting token- and sample-level signals, ensuring both data quality and diversity. The procedure consists of four steps.

**Step 1: MAB-Driven Cluster Selection.** We first partition the data pool $D$ into $k$ clusters $\{C_1, \ldots, C_k\}$. To avoid evaluating every sample exhaustively, we employ the Upper Confidence Bound (UCB) algorithm to prioritize clusters with the highest expected gain. At iteration $t$, the cluster score of $C_i$ and the selected cluster are defined as:

$$CS_i(t) = \bar{I}_i(t) + \gamma\sqrt{\frac{2\ln\left(\sum_{j=1}^k T(C_j, t)\right)}{T(C_i, t)}}, \quad C^*(t) = \arg\max_i CS_i(t), \tag{8}$$

where $\bar{I}_i(t)$ is the average influence score of samples in $C_i$ up to round $t$, $T(C_i, t)$ is the number of times $C_i$ has been sampled, and $\gamma$ balances exploration and exploitation.

**Step 2: Valuable Token Detection with Token-Level Utility.** Within the selected cluster $C^*(t)$, we compute token-level utility scores, including *Learning Gain (LG)* (Eq. 2) and *Answer Uncertainty (AU)* (Eq. 18). Tokens are then categorized as learnable, ambiguous, or uninformative using thresholds $(\tau_{LG}, \tau_{AU})$.

**Step 3: Sample Selection with Sample-Level Utility.** To connect token-level informativeness with sample-level data valuation, we compute the *Sample Utility Function* (Eq. 7). Only samples with the highest sample utility scores in $C^*(t)$ are selected for training, ensuring that retained samples provide maximal learning signal under a limited computational budget.

**Step 4: Cluster Score Updates for Next Round.** Once the subset is selected, we update the average influence score of cluster $C_i$ based in these samples as:

$$\bar{I}_i(t+1) = \frac{\bar{I}_i(t)\, T(C_i, t) + \sum_{x \in S_i} U_k^{\text{Sample}}(x)}{T(C_i, t) + |S_i|}, \tag{9}$$

where $S_i$ is the subset of selected samples from $C_i$ at iteration $t$. This update refines $CS_i(t)$, allowing the MAB scheduler to balance exploitation of high-quality clusters with exploration of under-visited ones in future rounds.

## 2.4 TOKEN-AWARE TRAINING WITH GATED OPTIMIZATION

The dual-level utility function not only enables efficient sample selection but also assigns each token a categorical label $\hat{y}_{i,j}$ (Eq. 5), indicating whether it is learnable ($\hat{y}_{i,j} = 1$), ambiguous ($\hat{y}_{i,j} = 2$), or uninformative ($\hat{y}_{i,j} = 0$). To exploit this decomposition during training, we propose a gated optimization strategy that routes tokens into distinct objectives based on their labels. This ensures that learnable tokens provide strong supervision, ambiguous tokens contribute through uncertainty-aware distillation, and uninformative tokens are suppressed to avoid noise amplification.

**Cross-Entropy Loss for Learnable Tokens.** Tokens labeled as $\hat{y}_{i,j} = 1$ are directly optimized with standard cross-entropy loss:

$$L_{\text{CE}} = \frac{1}{\sum_j \mathbf{1}[\hat{y}_{i,j} = 1]} \sum_{j=1}^{n_i} \mathbf{1}[\hat{y}_{i,j} = 1] \big( -\log p_\theta(y_{i,j} \mid x_i) \big), \tag{10}$$

ensuring that informative tokens continue to drive effective parameter updates.

**Self-Distillation Loss for Ambiguous Tokens.** Tokens labeled as $\hat{y}_{i,j} = 2$ are inherently ambiguous and thus optimized via masked self-distillation. Using a softened teacher distribution with temperature $T > 0$. Formally, let $z_\theta(x) \in \mathbb{R}^V$ be the logits, $p_\theta = \text{softmax}(z_\theta)$. For token position $(i,j)$, let the teacher give a distribution $q_{i,j} \in \Delta^{V-1}$ and the ground-truth token be $Y_{i,j} \sim q_{i,j}$ (multi-answer tokens correspond to high-entropy $q_{i,j}$). We compute the distillation loss as

$$L_{\text{SD}} = \frac{T^2}{\sum_j \mathbf{1}[\hat{y}_{i,j} = 2]} \sum_{j=1}^{n_i} \mathbf{1}[\hat{y}_{i,j} = 2] \left( -\sum_{v=1}^{V} q(v) \log p_\theta(v) + \text{const}(q) \right). \tag{11}$$

**Final Training Objective.** Uninformative tokens ($\hat{y}_{i,j} = 0$) are masked out and do not contribute to optimization. The overall objective combines cross-entropy and self-distillation with a balancing coefficient $\lambda \in [0, 1]$:

$$L = \lambda L_{\text{CE}} + (1 - \lambda) L_{\text{SD}}. \tag{12}$$

## 3 THEORETICAL ANALYSIS

### 3.1 WHY AU CAPTURES MULTI-ANSWER TOKENS

- Theorems A.1–A.3 show that when the model has *multiple strong* next-token candidates (evidence concentrated on $m \geq 2$ tokens and relatively evenly split), AU is provably large.
- Conversely, Propositions A.4–A.5 exclude the main confound: if all candidates are uniformly weak (small $s$), AU must be small, and if AU is large, the evidence must be spread across at least two candidates in non-negligible shares. Together, these results justify the use of AU as a detector of tokens with **multiple correct answers**.

### 3.2 WHY TOP-$k$ LG-DENSITY TOKENS MAXIMIZES THE SAMPLE UTILITY

We define the top-$k$ sample utility score $U_k^{\text{Sample}}(x_i)$ in Appendix A.6. By Proposition 3.2, $U_k^{\text{Sample}}(x_i)$ upper-bounds (and typically strictly improves over) the full-token ratio that also counts tokens with small or negative densities (e.g., high-AU but low-LG positions). Selecting the global top-$K_{\text{samples}}$ by $U_k^{\text{Sample}}(x_i)$ maximizes the expected loss reduction per unit baseline (training budget) under our additive approximation.

**Proposition 3.1** (Trimming low-density tokens increases sample utility). *Fix $i$ and a nonempty $S$. If there exists $j^\star \in S$ with $\rho_{i,j^\star} < U_i(S)$, then $U_i(S \setminus \{j^\star\}) > U_i(S)$. More generally, for any $T \subseteq S$ consisting only of indices with $\rho_{i,j} \geq U_i(S)$, one has $U_i(T) \geq U_i(S)$, with strict inequality if at least one strict inequality $\rho_{i,j} > U_i(S)$ holds in $T$.*

*Proof.* Write $A = \sum_{j \in S} \Delta \ell_{i,j}$ and $B = \sum_{j \in S} b_{i,j}$, so $U_i(S) = A/B$. For $j^\star$ we have $\Delta \ell_{i,j^\star} < (A/B)\, b_{i,j^\star}$. Then

$$U_i(S \setminus \{j^\star\}) = \frac{A - \Delta \ell_{i,j^\star}}{B - b_{i,j^\star}} \;>\; \frac{A - (A/B)\, b_{i,j^\star}}{B - b_{i,j^\star}} \;=\; \frac{A}{B} \;=\; U_i(S).$$

The extension to any $T$ that removes all indices with $\rho_{i,j} < U_i(S)$ follows by repeating argument. $\square$

**Proposition 3.2** (Top-$k$ by density maximizes sample utility at fixed budget). *Fix $k \in \{1, \ldots, n_i\}$. Among all $S \subseteq \{1, \ldots, n_i\}$ with $|S| = k$, $U_i(S)$ is maximized by taking the $k$ indices with the largest densities $\rho_{i,j} = \Delta \ell_{i,j}/b_{i,j}$. See Proposition A.8 in Appendix A for a detailed proof.*

*In particular,*

$$U_i(\text{top-}k\ \rho) \;\geq\; U_i(\{1, \ldots, n_i\}) \quad \text{(the full-token utility)}.$$

### 3.3 Why Knowledge Distillation Loss (KD) for High AU Tokens and Cross-Entropy Loss (CE) for High LG Tokens Can Perform Best?

Let $z_\theta(x) \in \mathbb{R}^V$ be the logits, $p_\theta = \text{softmax}(z_\theta)$. For token position $(i,j)$, let the teacher give a distribution $q_{i,j} \in \Delta^{V-1}$ and the ground-truth token be $Y_{i,j} \sim q_{i,j}$ (multi-answer tokens correspond to high-entropy $q_{i,j}$).

- (a) The CE gradient coincides with the KD gradient *in expectation*, but CE introduces additional sampling noise whose variance grows with how spread out $q$ is, whereas KD has no sampling noise; see Lemma A.9.
- (b) With a sufficiently small step size, a *smaller gradient covariance* implies a *larger* expected decrease in the underlying smooth risk; see Lemma A.10.

**Theorem 3.3** (AU-high tokens: KD yields strictly larger expected decrease than CE). *Fix $(i,j)$ and assume $J_\theta \neq 0$. Consider one SGD step on this token with either CE (using a hard label $Y \sim q_{i,j}$) or KD (using the full $q_{i,j}$). Under Lemma A.9 and Lemma A.10, for any step size $\eta \in (0, \frac{1}{L}]$,*

$$\mathbb{E}\big[\Delta R^{\text{KD}}\big] \;\geq\; \mathbb{E}\big[\Delta R^{\text{CE}}\big],$$

*with strict inequality whenever $q_{i,j}$ has positive entropy (i.e., $AU(x_{i,j}) > 0$). Here $\Delta R^{\text{KD}}$ and $\Delta R^{\text{CE}}$ denote the one-step decrease of the same risk $R$ under KD and CE updates, respectively.*

**Proposition 3.4** (LG-high tokens: CE is never worse than the all-CE baseline). *Let $E$ be the set of LG-high tokens and $A$ the set of AU-high tokens (disjoint). The **all-CE** baseline optimizes*

$$\mathcal{L}_{\text{all-CE}}(\theta) = \sum_{(i,j) \in E \cup A} L^{\text{CE}}(p_\theta, Y_{i,j}).$$

*Here **CE baseline** refers to the all–cross-entropy training scheme: for the given set of tokens, we always use hard labels and CE, with no KD anywhere.*

*The proposed mix uses CE on $E$ and KD on $A$:*

$$\mathcal{L}_{\text{mix}}(\theta) = \sum_{(i,j) \in E} L^{\text{CE}}(p_\theta, Y_{i,j}) + \lambda \sum_{(i,j) \in A} L^{\text{KD}}(p_\theta, q_{i,j}) \quad (\lambda > 0).$$

*Let $\Delta R^{\text{all-CE}}$ and $\Delta R^{\text{mix}}$ denote the one-step decrease of the underlying smooth risk $R(\theta)$ under the all-CE baseline and our mixed scheme, respectively, using the same step size $\eta \in (0, \frac{1}{L}]$. For tokens in $E$, both methods use CE and thus have identical per-step behavior. For tokens in $A$, by Theorem 3.3, the mixed scheme has no smaller and typically strictly larger expected loss decrease than the all-CE baseline. Therefore, per SGD step,*

$$\mathbb{E}\big[\Delta R^{\text{mix}}\big] \;\geq\; \mathbb{E}\big[\Delta R^{\text{all-CE}}\big],$$

*with strict inequality if $A$ contains at least one positive-entropy (AU-high) token.*

## 4 EXPERIMENT

### 4.1 EXPERIMENTAL SETUPS

**Datasets.** To investigate data selection across various scenarios and demonstrate the robustness of TOKENTUNE, we use two distinct data pools: **(1) Tulu3** (Lambert et al., 2024): A large-scale, real- world SFT dataset presented by Ai2, containing million-level records across a wide variety of subjects, including mathematics, programming, and user dialogues. **(2) Openhermes2.5**: A dataset with over 1 million data points, sourced from 16 distinct origins, including MetaMath (Yu et al., 2023), CamelAI (Li et al., 2023) and others.

**Benchmarks and Metrics.** To comprehensively evaluate the efficacy of TOKENTUNE, we evaluate TOKENTUNE on three leaderbords, including OpenLLM Leaderboards, ALpacaEval and MT-Bench. For OpenLLM Leaderboards, we adopt seven tasks, including TyDiQA (Clark et al., 2020), HellaSwag (Zellers et al., 2019), ARC-C (Clark et al., 2018), BoolQ (Clark et al., 2019), GSM8K (Cobbe et al., 2021), HumanEval (Chen et al., 2021) and LogiQA (Liu et al., 2020).

**Baselines.** We study several existing state-of-the-art methods as our baselines for data selection, including FULL DATA, RANDOM SELECTION (Xia et al., 2024b), TOKENCLEAN (Pang et al., 2025), RHO (Lin et al., 2024), IFD (Li et al., 2024b), INSTAG (Lu et al.), ENTROPY (Xia et al., 2024b), SELECTIT (Liu et al., 2024a), TOKEN LENGTH (TL) (Xia et al., 2024b), ZIP (Yin et al., 2024), CAR (Ge et al., 2024), DEITA (Liu et al., 2024b), LEAD (Lin et al., 2025).

**Implementation Details of TOKENTUNE.** We evaluate TOKENTUNE using three foundational models (LLAMA-3.1-8B, Mistral-7B and Qwen2-7B) and utilize Low-Rank Adaption (LoRA) Hu et al. (2022) for parameter-efficient fine-tuning. The maximum learning rate is set as $2 \times 10^{-5}$ with a linear decay schedule, and the batch size is 8. We also fix the maximum input sequence length to 2080. Models are trained for 1 epoch on 4 A800 GPUs.

### 4.2 EXP-1: OVERALL PERFORMANCE

We first evaluate the overall performance of TOKENTUNE against state-of-the-art baselines, using the same budget of $50K$ samples, corresponding to $5\%$ of the data pool. Results are reported on two representative datasets: Tulu3 (Table 1) and Openhermes (Table 4), and performance is evaluated on the OpenLLM Leaderboard, which include eight benchmarks.

**Exp-1.1: Overall Performance on Tulu3.** Table 1 reports results on LLaMA3.1-8B and Qwen2-7B. Overall, TOKENTUNE consistently surpasses strong baselines, confirming its effectiveness. *(1) Consistent Effectiveness across LLMs.* TOKENTUNE achieves robust improvements across different architectures. On LLaMA3.1-8B, it reaches 60.28, outperforming TOKENTUNE (58.27) and Deita (57.51) by +2.01 and +2.77, respectively. A similar trend holds on Qwen2-7B, where TOKENTUNE obtains 59.67, again surpassing both sample- and token-level baselines. These results demonstrate that TOKENTUNE consistently delivers gains regardless of backbone choice. *(2) Small Data, Big Gains.* Impressively, TOKENTUNE achieves these results using only 5% of the data pool, even outperforming the FULL DATA baseline (59.65). This challenges the assumption that more data is always better, showing instead that high-quality selection can unlock superior performance with far smaller subsets. *(3) Superior to State-of-the-art Baselines.* Although some baselines show strengths on specific tasks (*e.g.,* TokenClean on TyDiQA, Deita on BoolQ), they fall short on reasoning-heavy benchmarks such as GSM8K. In contrast, TOKENTUNE maintains consistently strong results across all tasks, with notable improvements of +3.45 on GSM8K and +2.77 on TyDiQA. This confirms the robustness and adaptability of TOKENTUNE in selecting high-utility data across diverse evaluation settings.

### 4.3 EXP-2: THE EFFICIENCY OF TOKENTUNE

We evaluate the efficiency of TOKENTUNE compared to baseline methods, with the results primarily shown in Figure 4 and Figure 5. TOKENTUNE significantly reduces both the inference and training times, thanks to a dual-level data selection strategy and the use of the MAB scheduler.

**Exp-2.1: Performance *vs.* Latency.** We compare performance and inference latency across different methods, as shown in Figure 4. TOKENTUNE (marked with a star) consistently achieves the best performance-latency trade-off, occupying the upper-left region of the plot. In addition, TOKENTUNE delivers up to **5× faster inference time** compared to the baselines while maintaining top performance on all the benchmarks. This confirms that TOKENTUNE not only outperforms

Table 1: Comparison of performance across different benchmarks on the Tulu3 dataset. Green highlights the remarkable improvements over the `Random` baseline.

| Type | Method | TyDiQA | HellaSwag | ARC-C | BoolQ | GSM8K | HumanEval | LogiQA | Avg. |
|---|---|---|---|---|---|---|---|---|---|
| | | | | | **Llama3.1-8B** | | | | |
| Base | Base | 22.80 | 59.92 | 50.82 | 82.18 | 50.31 | 69.28 | 26.51 | 51.69 |
| | Full Data | 54.31 | 58.57 | 49.18 | 83.48 | 66.08 | 80.34 | 25.58 | 59.65 |
| | Random | 46.96 | 60.14 | 51.34 | 82.80 | 51.98 | 73.75 | 27.91 | 56.41 |
| Token-Level | TokenClean | 52.91 | 61.82 | 54.00 | 82.18 | 51.00 | 77.31 | 28.22 | 58.21 |
| | RHO | 48.02 | 60.09 | 53.61 | 81.04 | 51.00 | 73.42 | 26.67 | 56.26 |
| Sample-Level | IFD | 38.55 | 60.35 | 49.87 | 82.21 | 57.04 | 71.60 | 26.36 | 55.14 |
| | ZIP | 52.32 | 60.11 | 50.65 | 82.83 | 53.83 | 70.02 | 24.81 | 56.37 |
| | Entropy | 45.72 | 60.35 | 49.87 | 82.21 | 57.04 | 75.02 | 26.36 | 56.65 |
| | Instag | 44.97 | 60.66 | 50.39 | 84.34 | 58.80 | 75.50 | 26.67 | 57.33 |
| | CaR | 46.01 | 59.97 | 50.73 | 83.91 | 54.06 | 73.98 | 27.44 | 56.59 |
| | TL | 48.76 | 60.53 | 50.39 | 82.52 | 51.99 | 73.12 | 27.44 | 56.39 |
| | SelectIT | 44.44 | 60.21 | 52.89 | 83.29 | 56.37 | 72.19 | 27.94 | 56.76 |
| | Deita | 44.81 | 60.74 | 52.11 | 82.86 | 57.35 | 74.60 | 30.08 | 57.51 |
| Dual-Level | TOKENTUNE (Ours) | 57.16 | 61.55 | 53.92 | 84.40 | 60.49 | 76.09 | 28.37 | **60.28** |
| | | | | | **Qwen2-7B** | | | | |
| Base | Random | 48.18 | 57.02 | 49.01 | 83.13 | 76.88 | 75.43 | 31.32 | 60.14 |
| Token-Level | TokenClean | 45.21 | 57.01 | 52.88 | 84.65 | 76.57 | 77.48 | 31.12 | 60.70 |
| | RHO | 44.18 | 56.82 | 50.07 | 81.89 | 75.42 | 76.94 | 30.01 | 59.33 |
| Sample-Level | ZIP | 50.31 | 59.02 | 48.49 | 85.21 | 76.03 | 75.27 | 30.85 | 60.74 |
| | Entropy | 50.01 | 57.43 | 45.14 | 83.84 | 76.92 | 74.18 | 29.96 | 59.64 |
| | Instag | 50.99 | 58.99 | 49.44 | 84.31 | 76.80 | 75.98 | 27.75 | 60.61 |
| | CaR | 50.33 | 58.55 | 46.17 | 83.01 | 77.87 | 75.31 | 30.70 | 60.28 |
| | TL | 43.84 | 59.04 | 43.76 | 83.60 | 77.11 | 76.14 | 31.01 | 59.21 |
| | SelectIT | 47.18 | 58.02 | 44.75 | 82.18 | 76.17 | 74.62 | 30.71 | 59.09 |
| | Deita | 46.72 | 59.17 | 49.01 | 83.76 | 78.25 | 77.01 | 33.02 | 60.99 |
| Dual-Level | TOKENTUNE (Ours) | 52.42 | 59.39 | 52.80 | 85.45 | 80.63 | 77.44 | 33.80 | **63.13** |

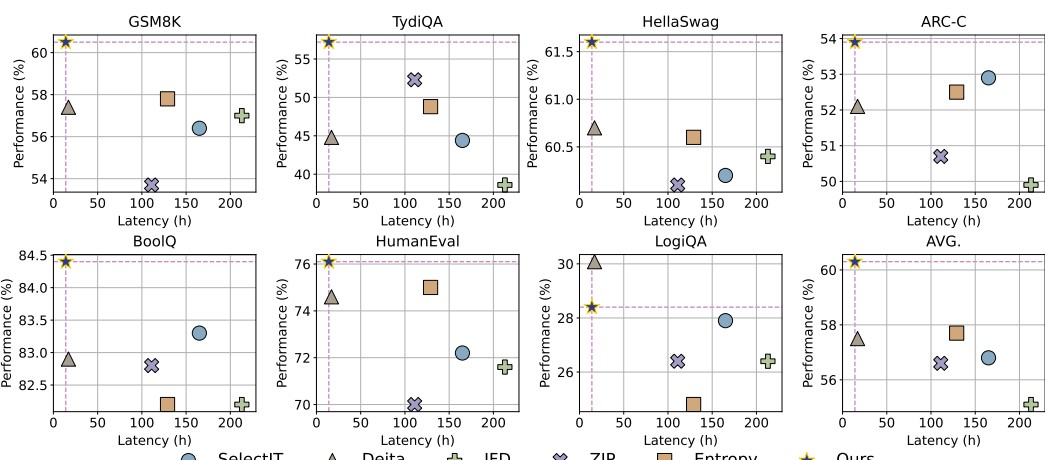

Figure 4: Comparison of Performance ($y$-axis) and Latency ($x$-axis) on different selection methods.

traditional methods in terms of model performance but also significantly reduces inference latency, making it a highly efficient solution for data selection and instruction tuning.

**Exp-2.2: Analysis of Latency Composition.** We evaluate the latency composition of TOKENTUNE by comparing inference time and training time across different data selection methods. The results, shown in Figure 5, reveal that TOKENTUNE reduces both inference and training times significantly, outperforming baseline methods. This demonstrates that TOKENTUNE effectively balances computational efficiency and model performance, making it a scalable solution for large-scale instruction tuning tasks.

Table 2: Ablation study of TokenTune across multiple benchmarks. "SD" denotes self-distillation loss, "TokUtility" denotes token-level influence score, "SamUtility" denotes sample-level influence score, and "MAB" denotes multi-armed bandit based cluster selection.

| Method | OpenLLM Leaderboards | | | | | | | Avg. |
|---|---|---|---|---|---|---|---|---|
| | TyDiQA | HellaSwag | ARC-C | BoolQ | GSM8K | HumanEval | LogiQA | |
| **TOKENTUNE (Ours)** | 57.16 | 61.55 | 53.92 | 84.40 | 60.49 | 76.09 | 28.37 | **60.28** |
| w/o SD | 58.43 | 61.17 | 52.37 | 82.98 | 56.13 | 75.65 | 27.31 | 59.06 |
| w/o MAB | 56.95 | 61.04 | 54.09 | 84.16 | 52.53 | 75.79 | 24.19 | 58.39 |
| w/o TokUtility | 53.79 | 59.77 | 48.66 | 83.57 | 54.82 | 74.23 | 26.67 | 57.36 |
| w/o SamUtility | 57.18 | 60.07 | 51.97 | 82.26 | 55.78 | 74.93 | 27.08 | 58.47 |

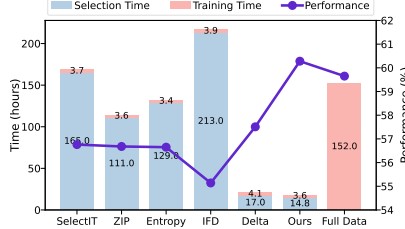

Figure 5: Inference Time (Full Data) and Training Time (Selected Data) per Iteration across Different Methods.

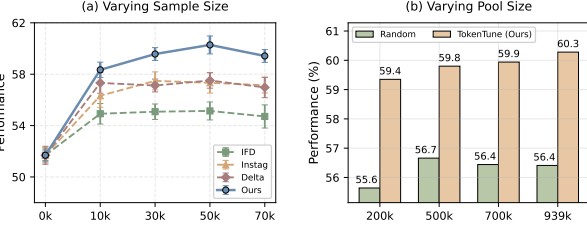

Figure 6: Avg Performance by Varying Sample Size and Pool Size.

## 4.4 Exp-3: Ablation Study of TokenTune

We investigate the effect of each component in TokenTune, with results shown in Table 2 and Figure 7. Removing any module leads to a clear performance drop: eliminating the token-level utility causes the largest degradation (–2.9), while excluding the sample-level utility, MAB scheduler, or self-distillation also results in noticeable declines. These results confirm that each component contributes to the overall effectiveness of TokenTune, and their integration is essential for achieving robust and consistent performance. More detailed results are provided in the Appendix F.

## 4.5 Exp-4: Analysis of Data Scaling

**Exp-4.1: Effect of Sample Size on Performance**. To examine the impact of data selection strategies on data scaling effectiveness, we conduc t experiments by selecting samples with varying budgets. As illustrated in Figure 6, TokenTune consistently presents higher average performance than alternative selection methods across all data quantities, achieving peak performance with only $50K$ samples. Notably, we observe a non-linear performance curve: gains taper and eventually decline beyond a certain data threshold, which reveals a crucial insight: "alignment-suitable data" is inherently limited. This finding challenges the conventional wisdom that more data automatically yields better results, underscoring the critical importance of strategic data selection over mere quantity. Please refer to Appendix **??** for more details.

**Exp-4.2: Effect of Pool Size on Performance.** We further examine how enlarging the candidate data pool affects the effectiveness of different selection strategies. As shown in Figure 6 (b), TokenTune consistently achieves higher performance than the random baseline across all pool sizes. Notably, its advantage becomes more pronounced as the pool expands: when moving from $200k$ to nearly 1M candidates, TokenTune steadily improves and reaches the best overall scores, while random selection shows only marginal gains and even plateaus. This demonstrates that TokenTune can effectively exploit larger pools to identify high-utility samples, confirming its scalability and robustness under data scaling. More detailed results are provided in the Appendix G.2.

## 5 Conclusion

In this paper, we present TokenTune, a dual-level data selection framework for instruction tuning that jointly considers token- and sample-level utilities. By capturing both learnable and uncertain tokens, TokenTune constructs an efficient utility function that avoids redundant inference while preserving diversity. The framework further integrates multi-armed bandit-based cluster selection with token-aware gated optimization, enabling scalable and effective training on large datasets. Experimental results demonstrate that TokenTune consistently outperforms state-of-the-art methods, achieving superior model performance with substantially less data and reduced training time.

## LLM USAGE

LLMs are used only for auxiliary purposes, such as language refinement, minor code debugging, synthetic data construction, and experiment evaluation support. They do not contribute to the research design, methodology, or core writing of the paper. Accordingly, LLM usage does not constitute a substantive contribution to the intellectual content of this work.

## ETHICS STATEMENT

All experiments in this paper are conducted on publicly available datasets, which contain no private, personal, or sensitive information. The proposed framework focuses on data selection and optimization strategies for instruction tuning, and does not involve generating or handling harmful or offensive content. By improving the efficiency and robustness of large-scale training, our method provides a general methodology that can be broadly applied to various language model applications without raising additional ethical risks. Nevertheless, as with all data-driven approaches, potential biases in the underlying datasets may propagate to downstream models, and careful auditing of training corpora remains an important future direction.

## REPRODUCIBILITY STATEMENT

We have made significant efforts to ensure the reproducibility of our work. The full implementation of our proposed method, including model training, evaluation scripts, and instructions for data construction, is publicly available at `https://anonymous.4open.science/r/TokenTune-D201/`. All experiments can be reproduced by following the provided scripts with the described hyperparameters. Details of implementation are included in the supplementary materials.

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

# Appendix Contents

# A  DETAILED THEORETICAL ANALYSIS

**AU Score**   We recall the definition of the AU score in a compact form. For a token position $(i, j)$ with target token $x_{i,j}$, let $\boldsymbol{\alpha}(x_{i,j}) = (\alpha_1(x_{i,j}), \dots, \alpha_K(x_{i,j}))$ denote the Dirichlet concentration parameters induced from the logits (see Eq. equation 18 in the main text), and write $\alpha_0(x_{i,j}) := \sum_{k=1}^{K} \alpha_k(x_{i,j})$. The answer uncertainty at $(i, j)$ is defined as the expected predictive entropy of a categorical parameter $\mathbf{p}$ drawn from this Dirichlet:

$$\mathrm{AU}(x_{i,j}) = \mathbb{E}_{\mathbf{p}\sim\mathrm{Dir}(\boldsymbol{\alpha}(x_{i,j}))}\Big[-\sum_{k=1}^{K} p_k \log p_k\Big] = -\sum_{k=1}^{K} \frac{\alpha_k(x_{i,j})}{\alpha_0(x_{i,j})}\Big(\psi(\alpha_k(x_{i,j})+1)-\psi(\alpha_0(x_{i,j})+1)\Big),$$

where $\psi(\cdot)$ is the digamma function. For brevity, in the following we drop the explicit dependence on $(i, j)$ and write $\alpha_k$ and $\alpha_0$ when no confusion arises.

For convenience write the total evidence $s := \alpha_0 > 0$ and the evidence shares $\beta_k := \alpha_k/\alpha_0$ (so $\sum_k \beta_k = 1$). Then equation 17 is equivalently

$$\mathrm{AU}(s, \boldsymbol{\beta}) = \psi(s+1) - \sum_{k=1}^{K} \beta_k\,\psi(s\beta_k+1). \tag{13}$$

## A.1  WHY AU CAPTURES MULTI-ANSWER TOKENS?

**Proof 1: High AU Implies the token has multiple correct answers**

- Theorems A.1–A.3 show that when the model has *multiple strong* next-token candidates (evidence concentrated on $m \geq 2$ tokens and relatively evenly split), AU is provably large.
- Conversely, Propositions A.4–A.5 exclude the main confound: if all candidates are uniformly weak (small $s$), AU must be small, and if AU is large, the evidence must be spread across at least two candidates in non-negligible shares. Together, these results justify the use of AU as a detector of tokens with **multiple correct answers**.

**Theorem A.1** (Sufficiency I: more strong candidates ⇒ larger AU)**.** *Fix $s > 0$. Suppose the evidence is concentrated on a set $S$ of size $m \geq 1$ and is evenly split: $\alpha_k = s/m$ for $k \in S$ and $\alpha_k = 0$ otherwise. Then*

$$\mathrm{AU}_m = \psi(s+1) - \psi\Big(\frac{s}{m}+1\Big),$$

*which is strictly increasing in $m$, with $\mathrm{AU}_1 = 0$ and $\mathrm{AU}_m > 0$ for $m \geq 2$.*

*Proof.* Under the hypothesis, $\beta_k = 1/m$ for $k \in S$ and 0 otherwise, and equation 13 gives the stated formula. Since $\psi$ is strictly increasing on $(0, \infty)$, $\psi(s/m+1)$ strictly decreases with $m$, hence $\mathrm{AU}_m$ strictly increases. The boundary values follow directly. $\square$

**Lemma A.2** (Strict concavity of the AU objective on the simplex interior)**.** *Fix $s > 0$ and let $F(\boldsymbol{\beta}) := \mathrm{AU}(s, \boldsymbol{\beta})$ restricted to $\sum_k \beta_k = 1$ and $\beta_k > 0$. Then $F$ is strictly concave on the interior of the simplex. In particular, the only stationary point is where all* active *components are equal.*

*Proof.* From equation 13, $F(\boldsymbol{\beta}) = \psi(s+1) - \sum_k \phi(\beta_k)$ with $\phi(x) := x\,\psi(sx+1)$. Thus $F''_{kk} = -\phi''(\beta_k)$ and $F''_{ij} = 0$ for $i \neq j$. Using polygamma notation $\psi^{(m)}$, we compute

$$\phi'(x) = \psi(sx+1) + sx\,\psi^{(1)}(sx+1), \quad \phi''(x) = 2s\,\psi^{(1)}(sx+1) + s^2 x\,\psi^{(2)}(sx+1).$$

For $x > 0$ we have $\psi^{(1)}(t) = \sum_{n=0}^{\infty} \frac{1}{(t+n)^2} > 0$ and $\psi^{(2)}(t) = -2\sum_{n=0}^{\infty} \frac{1}{(t+n)^3} < 0$. Therefore

$$2\psi^{(1)}(t) + t\,\psi^{(2)}(t) = 2\sum_{n=0}^{\infty}\Big(\frac{1}{(t+n)^2} - \frac{t}{(t+n)^3}\Big) = 2\sum_{n=1}^{\infty} \frac{n}{(t+n)^3} > 0,$$

so $\phi''(x) > 0$ and hence $F''_{kk} < 0$, proving strict concavity. Stationarity under the linear constraint $\sum_k \beta_k = 1$ yields $\phi'(\beta_k) = \lambda$ for all active $k$, and since $\phi'$ is strictly increasing, all active $\beta_k$ must be equal. $\square$

**Theorem A.3** (Sufficiency II: at fixed support size, AU is maximized by even split). *Fix $s > 0$ and an integer $m \geq 1$. Among all $\boldsymbol{\beta}$ supported on exactly $m$ indices, $\mathrm{AU}(s, \boldsymbol{\beta})$ attains its unique maximum at the uniform point $\beta_k = 1/m$ on the active set, with value $\psi(s+1) - \psi(s/m+1)$.*

This means: If exactly $m$ candidates are truly active, the best way (for AU) is to split evidence evenly across them; any imbalance lowers AU, and dropping to fewer than $m$ active candidates also lowers AU.

*Proof.* By Lemma A.2, $F$ is strictly concave on the simplex interior, so any interior stationary point is the unique global maximizer on that face. Lemma A.2 also shows all active coordinates must be equal at a stationary point. Evaluating at $\beta_k = 1/m$ gives the value in the claim. On the boundary (some $\beta_k \to 0$) the support size drops to $m' < m$, and Theorem A.1 then implies a strictly smaller maximum. $\qquad\square$

**Proposition A.4** (Necessary A: low total evidence cannot yield large AU). *Uniformly over $\boldsymbol{\beta}$, $\mathrm{AU}(s, \boldsymbol{\beta}) = O(s)$ as $s \to 0$. More precisely,*

$$\mathrm{AU}(s, \boldsymbol{\beta}) = \psi'(1)\, s \Big(1 - \sum_k \beta_k^2\Big) + O(s^2) \leq \frac{\pi^2}{6}\, s + O(s^2).$$

*Hence, if $\mathrm{AU} \geq \eta > 0$ then necessarily $s = \alpha_0 \geq \frac{6}{\pi^2}\, \eta\, (1 + o(1))$ as $\eta \downarrow 0$.*

*Proof.* Expand $\psi(1 + \varepsilon) = \psi(1) + \psi'(1)\varepsilon + O(\varepsilon^2)$ in equation 13 with $\varepsilon = s$ and $\varepsilon = s\beta_k$, respectively; $\psi(1)$ cancels and $\psi'(1) = \pi^2/6$. $\qquad\square$

**Proposition A.5** (Necessary B: large AU forces dispersion across candidates). *For any $s > 0$ and $\boldsymbol{\beta}$,*

$$\mathrm{AU}(s, \boldsymbol{\beta}) \;\geq\; \psi(s+1) - \psi\big(s \textstyle\sum_k \beta_k^2 + 1\big).$$

*Consequently, if $\mathrm{AU} \geq \eta > 0$, then*

$$\sum_k \beta_k^2 \;\leq\; r(s, \eta) := \frac{\psi^{-1}\big(\psi(s+1) - \eta\big) - 1}{s},$$

*and in particular the number of non-negligible candidates satisfies $\{k : \beta_k > 0\} \;\geq\; \lceil 1/r(s, \eta) \rceil$.*

*Proof.* Since $\psi$ is concave on $(0, \infty)$,

$$\sum_{k=1}^{K} \beta_k\, \psi(s\beta_k + 1) \;\overset{\substack{\text{(Jensen, concave } \psi)}}{\leq}\; \psi\left(\sum_{k=1}^{K} \beta_k\, (s\beta_k + 1)\right) \tag{14}$$

$$= \psi\left(s \sum_{k=1}^{K} \beta_k^2 + \sum_{k=1}^{K} \beta_k\right) \;=\; \psi\left(s \sum_{k=1}^{K} \beta_k^2 + 1\right). \tag{15}$$

Substitute into equation 13 to obtain the lower bound.

The stated upper bound on $\sum_k \beta_k^2$ then follows from the monotonicity of $\psi$ and its inverse. Finally, by Cauchy–Schwarz, $\sum_k \beta_k^2 \geq 1/m$ if at most $m$ components are nonzero, yielding the cardinality claim. $\qquad\square$

## A.2 WHY TOP-k LG-DENSITY TOKENS MAXIMIZES THE SAMPLE UTILITY

**Definition A.6** (Top-k Sample Utility Score). For each sample $x_i$, compute per-token LG $\Delta\ell_{i,j}$ and baseline $b_{i,j}$, form densities $\rho_{i,j} = \Delta\ell_{i,j}/b_{i,j}$ in Proposition 3.2, and define

$$U_k^{\text{Sample}}(x_i) := \frac{\sum_{j \in \text{top-}k(\rho)} \Delta\ell_{i,j}}{\sum_{j \in \text{top-}k(\rho)} b_{i,j}} = \frac{\sum_{j \in \text{top-}k(\rho)} \big(\ell_{\text{ref}}(x_{i,j}) - \ell_0(x_{i,j})\big)}{\sum_{j \in \text{top-}k(\rho)} \ell_0(x_{i,j})}.$$

By Proposition A.8, $U_k^{\text{Sample}}(x_i)$ upper-bounds (and typically strictly improves over) the full-token ratio that also counts tokens with small or negative densities (e.g., high-AU but low-LG positions). Selecting the global top-$K_{\text{samples}}$ by $U_k^{\text{Sample}}$ maximizes the expected loss reduction per unit baseline (training budget) under the additive approximation.

**Proposition A.7** (Trimming low-density tokens increases sample utility). *Fix $i$ and a nonempty $S$. If there exists $j^\star \in S$ with $\rho_{i,j^\star} < U_i(S)$, then $U_i(S \setminus \{j^\star\}) > U_i(S)$. More generally, for any $T \subseteq S$ consisting only of indices with $\rho_{i,j} \geq U_i(S)$, one has $U_i(T) \geq U_i(S)$, with strict inequality if at least one strict inequality $\rho_{i,j} > U_i(S)$ holds in $T$.*

*Proof.* Write $A = \sum_{j \in S} \Delta\ell_{i,j}$ and $B = \sum_{j \in S} b_{i,j}$, so $U_i(S) = A/B$. For $j^\star$ we have $\Delta\ell_{i,j^\star} < (A/B)\, b_{i,j^\star}$. Then

$$U_i(S \setminus \{j^\star\}) = \frac{A - \Delta\ell_{i,j^\star}}{B - b_{i,j^\star}} \;>\; \frac{A - (A/B)\, b_{i,j^\star}}{B - b_{i,j^\star}} \;=\; \frac{A}{B} \;=\; U_i(S).$$

The extension to any $T$ that removes all indices with $\rho_{i,j} < U_i(S)$ follows by repeating the argument. $\square$

**Proposition A.8** (Top-$k$ by density maximizes sample utility at fixed budget). *Fix $k \in \{1, \ldots, n_i\}$. Among all $S \subseteq \{1, \ldots, n_i\}$ with $|S| = k$, $U_i(S)$ is maximized by taking the $k$ indices with the largest densities $\rho_{i,j} = \Delta\ell_{i,j}/b_{i,j}$. In particular,*

$$U_i(\text{top-}k\ \rho) \;\geq\; U_i(\{1, \ldots, n_i\}) \quad \text{(the full-token utility).}$$

*Proof.* If $S$ is not the top-$k$ set, there exists $p \in S$ and $q \notin S$ with $\rho_{i,p} < \rho_{i,q}$. Consider $S' = (S \setminus \{p\}) \cup \{q\}$. Since $U_i(S)$ is a weighted average of $\{\rho_{i,j}\}_{j \in S}$, we have $U_i(S) \leq \max_{j \in S} \rho_{i,j} < \rho_{i,q}$. Replacing $p$ by $q$ strictly increases the average; iterating yields the top-$k$ set. The inequality $U_i(\text{top-}k) \geq U_i(\text{full})$ follows from Proposition A.7 by trimming all indices with $\rho_{i,j} < U_i(\text{full})$. $\square$

## A.3 WHY SELF-DISTILLATION LOSS FOR MULTI-ANSWER TOKENS CAN PERFORM BEST?

Let $z_\theta(x) \in \mathbb{R}^V$ be the logits, $p_\theta = \text{softmax}(z_\theta)$. For token position $(i, j)$, let the teacher give a distribution $q_{i,j} \in \Delta^{V-1}$ and the ground-truth token be $Y_{i,j} \sim q_{i,j}$ (multi-answer tokens correspond to high-entropy $q_{i,j}$). Define the two losses:

$$\underbrace{L^{\text{CE}}(p_\theta, Y) = -\log p_\theta(Y)}_{\text{cross-entropy}}, \qquad \underbrace{L^{\text{KD}}(p_\theta, q) = \text{KL}(q \,\|\, p_\theta) = -\sum_{v=1}^{V} q(v) \log p_\theta(v) + \text{const}(q).}_{\text{self-distillation}}$$

**Gradient (w.r.t. logits).** For softmax, $\nabla_z L^{\text{CE}}(p_\theta, Y) = p_\theta - e_Y$, $\nabla_z L^{\text{KD}}(p_\theta, q) = p_\theta - q$, where $e_Y$ is the one-hot of $Y$. For model parameters, $\nabla_\theta L = J_\theta^\top \nabla_z L$, $J_\theta = \frac{\partial z_\theta}{\partial \theta}$.

**Analysis**

- (a) CE's gradient equals KD's gradient *in expectation*, but CE adds extra noise that grows with how spread-out $q$ is; KD has no sampling noise. see details in Lemma A.9
- (b) With a small step size, *smaller gradient covariance* means a *larger* expected decrease. See details in Lemma A.10

**Lemma A.9** (Unbiasedness and variance of per-step gradients). *Conditioned on $x$ and $q$, for the random label $Y \sim q$,*

$$\mathbb{E}[\nabla_\theta L^{\text{CE}}(p_\theta, Y)] = \nabla_\theta L^{\text{KD}}(p_\theta, q), \quad \text{Cov}[\nabla_\theta L^{\text{CE}}(p_\theta, Y)] = J_\theta^\top \big(\text{Diag}(q) - qq^\top\big) J_\theta \;\succeq\; 0.$$

*In particular, the covariance is $0$ iff $q$ is a delta (entropy $= 0$).*

**Lemma A.10** (One-step expected progress under $L$-smooth risk). *Let $R(\theta)$ be an $L$-smooth objective and update $\theta^+ = \theta - \eta g$, where $g$ is an unbiased gradient estimator of $\nabla R(\theta)$. Then*

$$\mathbb{E}[R(\theta^+)] \;\leq\; R(\theta) - \eta \|\nabla R(\theta)\|^2 + \frac{L\eta^2}{2}\Big(\|\nabla R(\theta)\|^2 + \text{Tr}(\text{Cov}[g])\Big).$$

*Hence for a fixed small step $\eta$,* smaller gradient covariance *yields* larger *expected decrease.*

**Theorem A.11** (AU-high tokens: KD yields strictly larger expected decrease than CE). *Fix $(i,j)$ and assume $J_\theta \neq 0$. Consider one SGD step on this token with either CE (using a hard label $Y \sim q_{i,j}$) or KD (using the full $q_{i,j}$). Under Lemma A.9 and Lemma A.10, for any step size $\eta \in (0, \frac{1}{L}]$,*

$$\mathbb{E}\big[\Delta R_{\mathrm{KD}}\big] \ \geq \ \mathbb{E}\big[\Delta R_{\mathrm{CE}}\big],$$

*with strict inequality whenever $q_{i,j}$ has positive entropy (i.e., the AU of $(i,j)$ is nonzero). Here $\Delta R$ denotes the one-step drop of the underlying smooth risk.*

**Proposition A.12** (LG-high tokens: using CE is never worse than the CE-baseline). *Let $E$ be the set of LG-high tokens and $A$ the set of AU-high tokens (disjoint). The **all-CE** baseline optimizes $\sum_{(i,j)\in E\cup A} L^{\mathrm{CE}}(p_\theta, Y_{i,j})$. Here **CE-baseline** means the all-cross-entropy training scheme—i.e., for the given set of tokens, we use hard labels and CE everywhere, with no KD anywhere.*

*The proposed mix uses CE on $E$ and KD on $A$:*

$$\mathcal{L}_{\mathrm{mix}}(\theta) = \sum_{(i,j)\in E} L^{\mathrm{CE}}(p_\theta, Y_{i,j}) + \lambda \sum_{(i,j)\in A} L^{\mathrm{KD}}(p_\theta, q_{i,j}) \quad (\lambda > 0).$$

*For tokens in $E$, both methods use CE, thus identical per-step behavior. For tokens in $A$, by Theorem A.11, the mix has* no smaller *and typically* strictly larger *expected loss decrease than the all-CE baseline. Therefore, per SGD step, $\mathbb{E}[\Delta R_{\mathrm{mix}}] \geq \mathbb{E}[\Delta R_{\mathrm{all-CE}}]$, with strict inequality if $A$ contains at least one positive-entropy token.*

# B  RELATED WORK

**Data Selection for Instruction Tuning.** Previous works on data selection (Xia et al., 2024a; Zhou et al., 2023; Hanmo et al., 2024) can be broadly categorized into two key approaches: sample-level methods and token-level methods. Sample-level approaches rely on various metrics: perplexity-based selection (Marion et al., 2023; Li et al., 2024a) favors simpler patterns, diversity-aware methods (Wu et al., 2023; Yu et al., 2024) promote broad coverage but depend heavily on pretrained embeddings, quality-based metrics such as influence scoring (Xia et al., 2024a; Ghorbani & Zou, 2019; Kwon et al.; Choe et al., 2024) or external model evaluation (Li et al., 2024c) provide stronger theoretical grounding but incur high computational cost, complexity-driven selection (Li et al., 2024b; Liu et al., 2024b) risks including noisy or overly difficult samples, and uncertainty-based metrics (Han et al.; Liu et al., 2024a) are unstable due to loss landscape irregularities. Despite their differences, these methods all focus on entire samples, overlooking that token quality within the same example can vary substantially. To address this issue, token-level approaches such as TokenClean (Pang et al., 2025) attempt to filter noise tokens. However, they typically discard uncertain tokens altogether, which can lead to overfitting to spurious deterministic patterns.

# C  EXPERIMENTAL DETAILS

## C.1  BASELINES

We study several existing state-of-the-art methods as our baselines for data selection.

*(1) Full Data*: Train the model using the entire data pool.

*(2) Random Selection* (Xia et al., 2024b): Randomly selects training samples.

*(3) Instruction-Following Difficulty (IFD)* (Li et al., 2024b): Selects samples based on a complexity metric measuring instruction-following difficulty.

*(4) Perplexity (PPL)* (Li et al., 2024a): Prioritizes uncertain samples with high perplexity.

*(5) K-Center-Greedy (KCG)* (Sener & Savarese, 2018): Maximizes diversity by iteratively choosing the sample farthest from the current selection.

*(6) SelectIT* (Liu et al., 2024a): Selects samples via uncertainty-aware self-reflection during instruction tuning.

*(7) Token Length (TL)* (Xia et al., 2024b): Selects samples with the longest response lengths.

*(8) ZIP* (Yin et al., 2024): prompting a strong LLM to estimate and select samples based on quality, relevance, and complexity scores.

## C.2 EVALUATION METRICS AND BENCHMARKS

We evaluate our method on seven representative tasks aligned with the multi-task training pool but drawn from distinct distributions, reflecting key LLM capabilities.

## C.2 EVALUATION METRICS AND BENCHMARKS

We evaluate our method on seven representative tasks aligned with the multi-task training pool but drawn from distinct distributions, reflecting key LLM capabilities.

- **Code Generation.** We use HUMANEVAL (Chen et al., 2021) to evaluate the code-writing capabilities of LLMs. Performance is measured via the widely adopted `pass@10` metric.
- **Math Reasoning.** We use GSM8K (Cobbe et al., 2021) to evaluate the mathematical abilities of models. We adopt an 8-shot setting and evaluate performance using the *exact match accuracy* metric.
- **Cross-lingual Question Answering.** To assess multilingual understanding, we utilize the TyDiQA (Clark et al., 2020) dataset. We report *F1 scores* for passage selection and answer span extraction tasks.
- **Commonsense Reasoning.** We adopt BoolQ (Clark et al., 2019) to evaluate the model's ability to understand yes/no questions based on short passages. Accuracy is used as the evaluation metric.
- **Scientific QA.** We use ARC-C (Clark et al., 2018) to evaluate the ability to answer grade-school science questions that require reasoning over knowledge and context. We report accuracy.
- **Multi-choice QA.** We include HELLASWAG (Zellers et al., 2019) as a commonsense completion benchmark with minimal surface cues. Accuracy is used as the evaluation metric.
- **Logical Reasoning.** We use LoGiQA (Liu et al., 2020) to assess formal logical reasoning, which requires deductive inference beyond surface clues. Accuracy is reported.

## C.3 IMPLEMENTATION DETAILS OF TOKENTUNE

We evaluate TOKENTUNE using four foundational models (LLAMA-3.1-8B, LLAMA-3.2-3B, LLAMA-2-13B and Qwen2-7B) and utilize Low-Rank Adaption (LoRA) Hu et al. (2022) for parameter-efficient fine-tuning. The maximum learning rate is set as $2 \times 10^{-5}$ with a linear decay schedule, and the batch size is 8. We also fix the maximum input sequence length to 2080. Models are trained for 1 epoch on 4 A800 GPUs.

In the **preprocessing stage**, we compute sentence-level embeddings for all training samples using the pretrained encoder `BAAI/bge-base-en-v1.5`, and construct clusters using K-Means with cosine similarity. The number of clusters is set to 1000 by default.

In the **selection stage**, we select samples based on their estimated utilities using our dual-level scoring scheme. The sampling budget is fixed at 5%, resulting in approximately 50K selected samples out of 1M candidates. Cluster-level sampling is guided by a multi-armed bandit (MAB) scheduler using the UCB algorithm, where each arm corresponds to one cluster (*i.e.,* 1000 arms in total). The exploration parameter is set to $\gamma = 0.001$. For checkpoint selection, we evaluate on the development sets of target benchmarks and select models based on average validation performance (*e.g.,* accuracy or F1, depending on the task).

All key hyperparameters and their search ranges are summarized in Appendix C.4.

## C.4 SEARCH RANGES OF HYPERPARAMETERS

To support reproducibility, we list all key hyperparameters involved in both the selection and training stages of our framework. For each hyperparameter, we report the default value used in our main

Table 3: Search ranges and default values for all hyperparameters.

| Stage | Hyperparameter | Default | Search Range | Description |
|---|---|---|---|---|
| Selection | Token Utility Threshold | 0.6 | {0.4, 0.5, 0.6, 0.7} | Threshold for assigning tokens to types based on utility scores |
| | Bandit Algorithm | UCB | {UCB, Thompson, EXP3} | Strategy for MAB-based cluster selection |
| | Number of Clusters ($k$) | 1000 | {500, 1000, 1500} | Number of clusters used in sample selection |
| | Sampling Budget (%) | 5% | {2.5%, 5%, 10%} | Percentage of total sample budgets |
| | Exploration Rate ($\gamma$) | 0.001 | {1e-4, 1e-3, 1e-2} | Exploration coefficient in MAB (for UCB/EXP3) |
| Training | Fine-tuning Epochs | 1 | {1, 2, 3} | Number of training epochs for fine-tuning |
| | Learning Rate | 2e-5 | {1e-5, 2e-5, 5e-5} | Learning rate for optimizer |
| | Batch Size | 16 | {8, 16, 32} | Batch size per GPU |
| | Max Sequence Length | 2048 | {1024, 2048} | Maximum length of input sequences |
| | Logit Temperature (SD) | 1.0 | {0.7, 1.0, 1.3} | Temperature used in self-distillation predictions |

Table 4: Comparison of performance across different benchmarks on Openhermes dataset.

| Type | Method | TyDiQA | HellaSwag | ARC-C | BoolQ | GSM8K | HumanEval | LogiQA | Avg. |
|---|---|---|---|---|---|---|---|---|---|
| | | | | **Llama3.1-8B** | | | | | |
| Base | Base | 22.80 | 59.92 | 50.82 | 82.18 | 50.31 | 69.28 | 26.51 | 51.69 |
| | Random | 49.44 | 59.31 | 51.75 | 82.16 | 58.27 | 71.98 | 27.46 | 62.15 |
| Sample-Level | IFD | 41.97 | 60.21 | 52.45 | 82.89 | 52.45 | 69.80 | 27.75 | 55.36 |
| | ZIP | 47.82 | 60.33 | 53.32 | 84.84 | 52.76 | 71.39 | 27.91 | 56.91 |
| | Entropy | 51.45 | 60.41 | 50.04 | 83.17 | 56.74 | 72.49 | 26.05 | 57.19 |
| | Instag | 47.21 | 60.31 | 52.28 | 83.19 | 60.76 | 72.50 | 26.82 | 62.71 |
| | TL | 46.61 | 60.89 | 52.71 | 83.14 | 54.36 | 73.36 | 28.53 | 57.09 |
| | SelectIT | 49.15 | 60.33 | 52.49 | 83.94 | 60.03 | 70.79 | 27.91 | 57.81 |
| | Deita | 42.50 | 60.80 | 51.85 | 83.17 | 54.59 | 72.92 | 29.77 | 60.97 |
| Dual-Level | TOKENTUNE (Ours) | 53.56 | 61.49 | 52.63 | 85.05 | 65.62 | 74.69 | 28.37 | 65.51 |

experiments and the range considered in sensitivity studies or tuning. These values are summarized in Table 3.

# D    OVERALL PERFORMANCE ON OPENHERMES DATASET

## D.1    THE EFFECTIVENESS ON OPENHERMES DATASET

As illustrated in Table 4, TOKENTUNE demonstrates impressive performance on the Openhermes dataset as well, surpassing all state-of-the-art baselines. On the LLaMA3.1-8B model, TOKENTUNE achieves an average score of 65.51, outpacing the best baseline, Deita (60.97), by a significant margin of +4.54. While some baselines such as SelectIT excel in specific tasks like PPL on Qwen2-7B, TOKENTUNE consistently maintains top-tier performance across various benchmarks. Notably, on the challenging HumanEval benchmark for code generation, TOKENTUNE demonstrates superior robustness, achieving higher performance than all other methods. These results further confirm TOKENTUNE 's ability to perform well across diverse models and benchmarks, emphasizing its consistent and scalable effectiveness for data selection and instruction tuning.

# E    ADDITIONAL BACKBONE ON TULU3 DATASET

A central question for data selection methods is whether the observed gains persist when scaling the backbone model size up or down. To assess the robustness and generality of our approach, we conduct additional experiments on two representative LLaMA-family models: a smaller model (LLaMA-3.2-3B) and a larger one (LLaMA-2-13B). All fine-tuning and evaluation settings follow the main experimental protocol.

## E.1    OVERALL PERFORMANCE ON LLAMA-3.2-3B

Table 5 reports the performance on seven benchmarks when finetuning LLaMA-3.2-3B with different data selection strategies. Despite the reduced capacity of the 3B model, our method achieves the highest average score (48.84) and consistently outperforms strong baselines such as DS2, Deita, ZIP, and TL across diverse tasks including TyDiQA, BoolQ, GSM8K, and HumanEval.

Table 5: Comparison of data selection methods on LLaMA3.2-3B across multiple benchmarks.

| Method | Benchmark | | | | | | | Average |
|--------|----------|------------|-------|-------|-------|-----------|--------|---------|
|        | TyDiQA | HellaSwag | ARC-C | BoolQ | GSM8K | HumanEval | LogiQA | |
| Base   | 32.67 | 55.10 | 42.20 | 73.01 | 28.48 | 50.11 | 22.17 | 43.39 |
| Random | 38.96 | 55.21 | 41.95 | 73.68 | 29.01 | 57.13 | 24.51 | 45.78 |
| DS2    | 39.96 | 55.31 | 44.27 | 74.86 | 29.94 | 55.71 | 23.72 | 46.25 |
| Deita  | 31.07 | 55.74 | 42.89 | 75.60 | 32.47 | 56.90 | 24.34 | 45.57 |
| ZIP    | 41.60 | 55.05 | 41.69 | 75.79 | 28.02 | 53.08 | 23.26 | 45.50 |
| Entropy| 40.01 | 54.37 | 41.12 | 74.36 | 28.62 | 55.62 | 23.91 | 45.43 |
| Instag | 37.02 | 55.40 | 42.46 | 72.85 | 30.76 | 56.17 | 24.50 | 45.59 |
| CaR    | 38.87 | 54.99 | 41.95 | 76.28 | 30.93 | 56.63 | 25.89 | 46.51 |
| TL     | 39.87 | 54.73 | 42.62 | 74.09 | 29.01 | 57.01 | 23.98 | 45.90 |
| ours   | 49.13 | 55.55 | 43.58 | 77.95 | 32.39 | 59.06 | 24.19 | **48.84** |

Table 6: Comparison of data selection methods on LLaMA-2-13B across multiple benchmarks.

| Method | Benchmark | | | | | | | Average |
|--------|----------|------------|-------|-------|-------|-----------|--------|---------|
|        | TyDiQA | HellaSwag | ARC-C | BoolQ | GSM8K | HumanEval | LogiQA | |
| Base   | 31.20 | 60.01 | 47.72 | 80.91 | 24.12 | 31.22 | 25.89 | 43.01 |
| Random | 35.96 | 60.29 | 48.75 | 82.09 | 32.01 | 39.61 | 28.68 | 46.77 |
| DS2    | 36.11 | 60.36 | 50.90 | 81.93 | 21.36 | 38.37 | 25.43 | 44.92 |
| Deita  | 34.86 | 60.95 | 49.01 | 82.21 | 28.87 | 40.88 | 26.82 | 46.23 |
| ZIP    | 40.08 | 60.21 | 50.65 | 82.80 | 26.19 | 36.49 | 27.13 | 46.22 |
| Entropy| 42.37 | 60.63 | 48.14 | 81.99 | 27.48 | 37.82 | 26.93 | 46.48 |
| Instag | 37.03 | 60.46 | 48.92 | 82.06 | 28.43 | 39.04 | 27.91 | 46.26 |
| CaR    | 37.55 | 60.22 | 47.98 | 82.06 | 30.55 | 44.57 | 31.63 | 47.79 |
| TL     | 41.26 | 60.47 | 49.53 | 82.37 | 34.16 | 34.16 | 28.06 | 46.52 |
| ours   | 44.86 | 60.79 | 50.71 | 82.01 | 31.73 | 44.02 | 28.07 | **48.88** |

These results demonstrate that our selection mechanism remains highly effective even in the low-capacity regime, highlighting the robustness of our dual-level selection strategy. The consistent gains also suggest that self-distillation on multi-answer tokens enhances generalization, enabling the model to make better use of informative supervision even under limited capacity.

### E.2 OVERALL PERFORMANCE ON LLAMA-2-13B

In this experiment, we present the results for LLaMA-2-13B, a substantially larger and more capable model. Table 6 shows the results on LLaMA-2-13B across seven benchmarks. Our method achieves the best performance across all reported tasks, consistently outperforming strong baselines such as DS2, Deita, ZIP, and TL. The improvements are particularly pronounced on reasoning and knowledge-intensive tasks such as ARC-C, BoolQ, and GSM8K. This further confirms that the proposed design principles generalize effectively across the model-scaling spectrum, from small to large backbones.

## F  DETAILED ABLATION STUDY

### F.1 ABLATION STUDY OF TOKENTUNE COMPONENTS.

We conduct a detailed ablation study to examine the contribution of each component in TOKENTUNE, with results reported in Table 2 and Figure 7. The findings show that removing any module leads to a noticeable degradation in performance. In particular, discarding the token-level utility causes the largest drop (–2.9 on average), highlighting its central role in identifying informative tokens. Similarly, eliminating the sample-level utility, multi-armed bandit scheduler, or self-distillation loss also results in consistent declines. These results confirm that all components are indispensable and that their integration is crucial for achieving robust improvements across diverse benchmarks.

Table 7: Performance of different replacement strategies in the Token Utility module.

| Module | Replace Strategy | Benchmark Performance | | | | | | | Avg. |
|---|---|---|---|---|---|---|---|---|---|
| | | TyDiQA | HellaSwag | ARC-C | BoolQ | GSM8K | HumanEval | LogiQA | |
| Token Utility (LG+AU) | LG+AU | 57.16 | 61.55 | 53.92 | 84.40 | 60.49 | 76.09 | 28.37 | **60.28** |
| | LG + Entropy | 57.01 | 61.58 | 52.37 | 83.76 | 57.73 | 74.14 | 25.58 | 58.88 |
| | LDP+ AU | 56.72 | 60.17 | 52.02 | 83.79 | 57.16 | 73.94 | 26.19 | 58.57 |
| | LDP + Entropy | 48.14 | 61.09 | 52.41 | 82.42 | 55.79 | 74.01 | 24.91 | 56.97 |

Table 8: Performance of Sample Utility with/without normalization.

| Module | Replace Strategy | Benchmark Performance | | | | | | | Avg. |
|---|---|---|---|---|---|---|---|---|---|
| | | TyDiQA | HellaSwag | ARC-C | BoolQ | GSM8K | HumanEval | LogiQA | |
| Sample Utility | Norm. | 57.16 | 61.55 | 53.92 | 84.40 | 60.49 | 76.09 | 28.37 | **60.28** |
| | w/o Norm. | 57.15 | 61.47 | 51.97 | 83.66 | 55.44 | 72.43 | 25.58 | 58.24 |

## F.2 The Effectiveness of Token-level Utility Function

To validate the design choices in our Token-Level Utility Function, we conduct a controlled ablation study comparing our full method (LG + AU) against several alternative strategies: (1) replacing LG with loss delta under noisy perturbations (LDP), and (2) replacing AU with Entropy. The results are shown in Table 7.

**LG Compared to LDP.** We replace LG with Noisy Loss (i.e., loss delta under perturbations) while keeping the AU component fixed. As shown in Table 6, this substitution consistently degrades performance across all benchmarks (e.g., average score drops from 60.28 to 58.57). This supports our design choice of using LG, which estimates the expected utility of tokens more efficiently and reliably. Unlike LDP, which primarily captures sensitivity to input noise and lacks awareness of the model's current learning dynamics, LG reflects the model's evolving uncertainty in a forward-only manner. It effectively prioritizes tokens that are expected to provide the most generalizable learning signal, without incurring the high computational cost of adversarial perturbations.

**AU Compared to Entropy.** We also evaluate the effect of replacing AU with standard entropy, keeping the LG component fixed. This substitution again leads to a noticeable performance drop (from 60.28 to 58.88), confirming the unique advantages of AU. This is because while entropy measures the flatness of the output distribution, it fails to distinguish between true ambiguity and model uncertainty due to low confidence. In contrast, AU explicitly targets multi-answer positions, which refer to tokens where the model assigns high probability to multiple plausible continuations, thereby capturing a semantically meaningful form of ambiguity. As a result, AU more precisely identifies tokens suitable for self-distillation, enhancing generalization during fine-tuning.

## F.3 The Effectiveness of Sample-level Utility Function

To evaluate the role of normalization in sample-level utility estimation, we conduct an ablation study comparing two variants: one with score normalization and one without. The normalization procedure adjusts raw sample utility scores to eliminate biases introduced by sequence length. As shown in Table 8, removing normalization results in a consistent performance drop across all benchmarks, with the average score declining from 60.28 to 58.24. These results further confirm that normalization plays a crucial role in mitigating length-induced bias. It ensures that the selection process emphasizes samples that are dense in learning signal, rather than those that are simply shorter or easier to fit, thereby enhancing the overall effectiveness of fine-tuning.

## F.4 The Effectiveness of MAB Module

To assess the contribution of the Multi-Armed Bandit (MAB) module in our data selection pipeline, we conduct an ablation comparing three widely used bandit algorithms: UCB, Thompson Sampling, and EXP3. The goal is to examine whether the observed performance gains are specific to the choice of bandit strategy or stem from the general idea of bandit-based adaptive selection.

As shown in Table 9, all three methods yield similar overall performance, with average scores ranging from 60.00 to 60.28. This indicates that our improvements are not tied to a particular bandit algorithm.

Table 9: Comparison of different MAB algorithms and clustering methods.

| Module | Replace Strategy | Benchmark Performance | | | | | | | Avg. |
|---|---|---|---|---|---|---|---|---|---|
| | | TyDiQA | HellaSwag | ARC-C | BoolQ | GSM8K | HumanEval | LogiQA | |
| MAB | UCB | 57.16 | 61.55 | 53.92 | 84.40 | 60.49 | 76.09 | 28.37 | 60.28 |
| | Thompson | 57.21 | 61.47 | 53.86 | 83.98 | 58.97 | 77.12 | 27.41 | 60.00 |
| | EXP3 | 58.19 | 61.52 | 53.92 | 83.97 | 59.18 | 76.09 | 27.62 | 60.07 |
| Cluster | Kmeans | 57.16 | 61.55 | 53.92 | 84.40 | 60.49 | 76.09 | 28.37 | 60.28 |
| | DBSCAN | 57.93 | 61.07 | 54.17 | 83.26 | 57.82 | 76.48 | 27.95 | 59.81 |
| | GMM | 56.14 | 61.53 | 54.89 | 84.23 | 59.25 | 76.92 | 28.30 | 60.18 |

Table 10: Ablation on self-distillation loss (SD) in token-level training.

| Module | Replace Strategy | Benchmark Performance | | | | | | | Avg. |
|---|---|---|---|---|---|---|---|---|---|
| | | TyDiQA | HellaSwag | ARC-C | BoolQ | GSM8K | HumanEval | LogiQA | |
| SD Loss | CE (LG) + SD (AU) | 57.16 | 61.55 | 53.92 | 84.40 | 60.49 | 76.09 | 28.37 | 60.28 |
| | CE (LG+AU) | 57.13 | 61.32 | 52.19 | 82.47 | 55.72 | 75.42 | 26.81 | 58.72 |

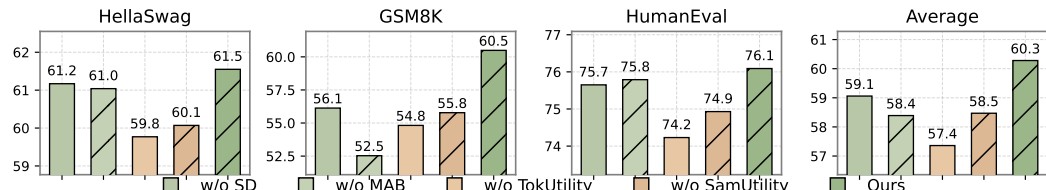

Figure 7: Ablation Study of TokenTune.

Instead, the key advantage lies in leveraging the exploration–exploitation paradigm to dynamically prioritize high-utility regions during sample selection. These results validate the robustness and generality of the MAB module design. The use of bandit-based control helps reduce redundant computation on low-reward areas and accelerates selection without relying on fine-grained tuning of the underlying algorithm.

### F.5 THE EFFECTIVENESS OF SELF-DISTILLATION LOSS

To investigate the effect of the self-distillation loss used for multi-answer tokens, we first conduct an ablation study where the self-distillation (SD) objective is replaced with standard cross-entropy (CE) loss. This variant removes the distinction between token types and treats all tokens as learnable. As shown in Table 10, removing self-distillation consistently degrades performance across benchmarks, especially on HumanEval and GSM8K, with the average score dropping from 60.3 to 58.7. These results demonstrate that self-distillation plays a critical role in handling multi-answer tokens, which often admit multiple plausible next-token candidates. Rather than forcing the model to commit to one specific label using CE loss, self-distillation encourages the model to maintain and refine its own distribution over plausible answers, allowing it to better generalize under ambiguous supervision.

## G DETAILED ANALYSIS OF DATA SCALING

### G.1 DETAILED PERFORMANCE OF VARYING SAMPLE SIZE

To further study scaling behavior under different training budgets, we evaluate TokenTune and baselines with varying sample sizes, as shown in Table 4. TokenTune consistently achieves superior performance across all data budgets, and its advantage is especially evident in low-data regimes (*e.g.,* +3.4 points over the best baseline at $10k$). Importantly, performance peaks around $50k$ samples, after which additional data yields diminishing or even negative returns. This non-linear trend suggests that alignment-suitable data is inherently limited, and emphasizes that quality-aware selection is more critical than sheer quantity.

Table 11: Performance comparison across different sample sizes.

| Method | TyDiQA | HellaSwag | ARC-C | BoolQ | GSM8K | HumanEval | LogiQA | Average |
|--------|--------|-----------|-------|-------|-------|-----------|--------|---------|
| **Base Model** | | | | | | | | |
| Base | 22.80 | 59.92 | 50.82 | 82.18 | 50.31 | 69.28 | 26.51 | 51.69 |
| **Sample Budget Ratio: 1%** | | | | | | | | |
| IFD | 39.79 | 60.38 | 51.59 | 81.69 | 51.53 | 72.34 | 27.13 | 54.92 |
| Instag | 42.37 | 60.27 | 51.77 | 83.17 | 54.59 | 75.23 | 26.82 | 56.32 |
| Deita | 46.56 | 60.65 | 52.37 | 83.20 | 56.66 | 72.19 | 29.61 | 57.32 |
| **Ours** | 51.66 | 61.56 | 54.61 | 84.43 | 54.13 | 74.72 | 27.29 | **58.34** |
| **Sample Budget Ratio: 2.5%** | | | | | | | | |
| IFD | 40.93 | 60.47 | 51.59 | 82.09 | 50.38 | 71.73 | 26.82 | 54.86 |
| Instag | 44.78 | 60.22 | 51.68 | 83.20 | 57.66 | 73.20 | 28.22 | 56.99 |
| Deita | 45.60 | 61.00 | 51.08 | 82.12 | 58.26 | 72.60 | 28.68 | 57.05 |
| **Ours** | **55.78** | **61.03** | **53.74** | **83.94** | **56.28** | **75.13** | **28.01** | **59.13** |
| **Sample Budget Ratio: 5%** | | | | | | | | |
| IFD | 38.55 | 60.35 | 49.87 | 82.21 | 57.04 | 71.60 | 26.36 | 55.14 |
| Instag | 44.97 | 60.66 | 50.39 | 84.34 | 58.80 | 75.50 | 26.67 | 57.33 |
| Deita | 44.81 | 60.74 | 52.11 | 82.86 | 57.35 | 74.60 | 30.08 | 57.51 |
| **Ours** | **57.16** | **61.55** | **53.92** | **84.40** | **60.49** | **76.09** | **28.37** | **60.28** |
| **Sample Budget Ratio: 7%** | | | | | | | | |
| IFD | 38.48 | 60.20 | 49.35 | 82.09 | 55.28 | 72.30 | 25.27 | 54.71 |
| Instag | 43.04 | 60.53 | 51.51 | 83.94 | 58.96 | 75.35 | 26.67 | 57.14 |
| Deita | 42.10 | 60.70 | 51.42 | 83.54 | 60.03 | 71.41 | 29.61 | 56.97 |
| **Ours** | 54.10 | 61.64 | 54.44 | 85.89 | 57.89 | 74.36 | 27.60 | **59.42** |
| **Sample Budget Ratio: 10%** | | | | | | | | |
| IFD | 39.48 | 60.18 | 50.73 | 82.15 | 55.44 | 73.41 | 26.82 | 55.46 |
| Instag | 44.17 | 60.02 | 52.54 | 83.17 | 56.13 | 72.19 | 28.06 | 56.61 |
| Deita | 44.57 | 60.80 | 51.94 | 83.29 | 59.80 | 72.46 | 28.99 | 57.41 |
| **Ours** | **59.15** | **61.13** | **53.40** | **83.89** | **57.20** | **76.04** | **26.82** | **59.66** |

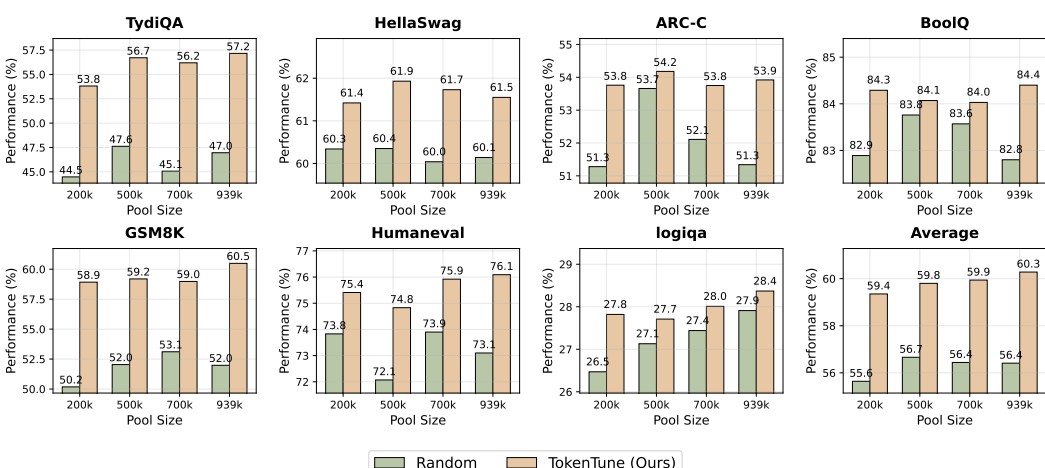

Figure 8: Performance by Varying Pool Size.

## G.2 DETAILED PERFORMANCE OF VARYING POOL SIZE

We also examine the effect of enlarging the candidate data pool, with results shown in Figure 6. TOKENTUNE consistently surpasses the random baseline under all pool sizes, and the margin grows larger as the pool expands. For example, when increasing from $200k$ to nearly $1M$ candidates, TOKENTUNE steadily improves and achieves the highest overall scores, whereas random selection

Table 12: PPerformance of different selection methods where utility scores are computed using LLaMA-3.1-8B and downstream fine-tuning is performed on Qwen2-7B.

| Selection Method | Benchmark | | | | | | | Average |
|---|---|---|---|---|---|---|---|---|
| (Backbone: LLaMA3.1-8B) | TyDiQA | HellaSwag | ARC-C | BoolQ | GSM8K | HumanEval | LogiQA | |
| IFD | 47.88 | 58.96 | 47.03 | 83.91 | 78.56 | 76.43 | 30.23 | 60.43 |
| SelectIT | 50.98 | 58.00 | 49.78 | 83.43 | 76.79 | 77.49 | 29.15 | 60.80 |
| Entropy | 46.19 | 58.48 | 45.56 | 85.27 | 76.92 | 76.96 | 28.99 | 59.77 |
| TL | 48.66 | 59.07 | 48.75 | 84.47 | 77.72 | 75.26 | 30.71 | 60.66 |
| Ours | 49.31 | 59.80 | 52.89 | 84.50 | 78.42 | 78.24 | 30.70 | **61.98** |

Table 13: Performance of different selection methods where utility scores are computed using GPT2 and downstream fine-tuning is performed on LLaMA-3.1-8B.

| Selection Method | Benchmark | | | | | | | Average |
|---|---|---|---|---|---|---|---|---|
| (Backbone: GPT2) | TyDiQA | HellaSwag | ARC-C | BoolQ | GSM8K | HumanEval | LogiQA | |
| IFD | 43.57 | 60.49 | 52.89 | 83.26 | 55.28 | 74.07 | 26.06 | 56.52 |
| SelectIT | 42.73 | 60.04 | 53.18 | 83.14 | 52.32 | 73.92 | 26.35 | 55.95 |
| Entropy | 43.51 | 60.37 | 51.94 | 84.47 | 51.15 | 71.52 | 25.89 | 55.55 |
| TL | 47.48 | 60.51 | 50.34 | 82.49 | 51.32 | 73.43 | 26.74 | 56.04 |
| Ours | 55.83 | 61.31 | 53.14 | 83.97 | 57.81 | 75.89 | 27.99 | **59.42** |

quickly plateaus. These results demonstrate that TOKENTUNE effectively leverages larger candidate pools to extract high-utility subsets, confirming its scalability and robustness in large-scale scenarios.

## H    THE GENERALIZATION OF TOKENTUNE

### H.1    PERFORMANCE ON CROSS-ARCHITECTURE SETTING

To evaluate the generality of our selection strategy across model architectures, we conduct experiments under a cross-family setting, where the model used to compute utility scores differs from the one used for downstream fine-tuning. Specifically, we compute token-level utilities using LLaMA-3.1-8B, while the selected data is used to fine-tune Qwen2-7B, a model from a different architecture.

As shown in Table 12, our method achieves the best average performance (61.98) across all baselines, outperforming strong methods such as SelectIT (60.80), and IFD (60.43). The improvement is consistent across most benchmarks, particularly on ARC-C (+3.11 over TL) and LogiQA (+2.01 over SelectIT), demonstrating the robustness of our utility estimation even when computed from a mismatched backbone. These results provide strong evidence that our utility scoring mechanism captures transferable signals of data quality that are not tied to a specific model architecture, validating its applicability in realistic settings where the scoring model and fine-tuning model may differ.

### H.2    PERFORMANCE ON CROSS-SCALE SETTING

To assess whether TOKENTUNE's utility estimation generalizes across model sizes, we evaluate its performance under a cross-scale setting, where the model used to compute utility scores is smaller than the model used for downstream fine-tuning. Specifically, token-level utilities are computed using a GPT2 model, while the selected data is used to fine-tune a larger Qwen2-7B model.

As shown in Table 13, TOKENTUNE maintains strong performance in this challenging setup, outperforming all baselines with an average score of 62.09, compared to 60.87 for SelectIT and 60.12 for IFD. The gains are particularly notable on ARC-C (+1.90 over TL) and LogiQA (+2.07 over SelectIT), indicating that the utility scores produced by a small model remain effective for guiding the fine-tuning of larger models. This demonstrates TOKENTUNE's ability to capture scale-invariant utility signals, validating its applicability in scenarios where computational budget limits the scoring model size.

Table 14: Effect of different token utility thresholds on performance.

| Threshold | Benchmark Performance | | | | | | | Average |
|---|---|---|---|---|---|---|---|---|
| | TyDiQA | HellaSwag | ARC-C | BoolQ | GSM8K | HumanEval | LogiQA | |
| 0.7 | 57.78 | 61.56 | 52.54 | 84.00 | 58.04 | 74.54 | 26.82 | 59.33 |
| 0.6 | 57.16 | 61.55 | 53.92 | 84.40 | 60.49 | 76.09 | 28.37 | **60.28** |
| 0.5 | 57.50 | 61.69 | 53.57 | 84.03 | 57.81 | 75.93 | 27.13 | 59.67 |
| 0.4 | 55.88 | 61.45 | 52.89 | 84.37 | 57.50 | 73.52 | 24.65 | 58.61 |

## I    PARAMETER SENSITIVITY ANALYSIS

### I.1    EFFECT OF THRESHOLDS FOR TOKEN UTILITY

We analyze the impact of varying the token utility threshold, which determines how tokens are assigned to different training objectives.

As shown in Table 14, threshold choice significantly affects performance. When the threshold is too high (*e.g.,* 0.7), the model includes noisy or uninformative tokens, which weakens supervision and degrades performance. Conversely, a threshold that is too low (*e.g.,* 0.4) filters out many informative tokens that still carry valuable learning signals, leading to under-utilization of training data. The best performance is achieved at a moderate threshold of 0.6, with an average score of 60.28. Notably, performance remains relatively stable within the range of 0.5 to 0.7, suggesting that the method is robust to small variations in threshold, as long as extreme values are avoided.

### I.2    EFFECT OF EXPLORATION RATE $\gamma$ OF MAB.

Our approach employs $\gamma$ to balance the diversity and quality during cluster sampling. As shown in Table 15, when $\gamma$ is small, the MAB framework prioritizes high-influence clusters and risks local optima due to reduced diversity. Conversely, when $\gamma$ is large, it overemphasizes diversity at the expense of quality, limiting model performance gains.

Table 15: Performance with varying exploration rate $\gamma$.

| Exploration Rate $\gamma$ | 0.0005 | 0.001 | 0.005 | 0.01 | 0.05 |
|---|---|---|---|---|---|
| Performance (Avg.) | 59.02 | 60.28 | 59.67 | 59.04 | 57.88 |

### I.3    EFFECT OF THE NUMBER OF CLUSTERS/ MAB ARMS $k$.

In our setup, each arm corresponds to a cluster. We use the Elbow method to guide the choice of arms/clusters $k$, which eliminates the need for manual adjustment. The result in Table 16 show that too few clusters (e.g., $k$=100) lead to high variance and under-representation of data, while too many clusters (e.g., $k$=5000) introduce redundancy and reduce exploration efficiency. A moderate choice of $k$=1000 provides the best balance between selection diversity and computational efficiency.

Table 16: Performance with varying number of MAB arms $K$.

| Number of MAB Arms $K$ | 100 | 500 | 1000 | 2000 | 3000 | 5000 |
|---|---|---|---|---|---|---|
| Performance (Avg.) | 58.73 | 58.91 | 60.28 | 60.01 | 59.67 | 59.04 |

### I.4    EFFECT OF DIFFERENT CLUSTERING ALGORITHM OF TOKENTUNE.

We compared Agglomerative Clustering, DBSCAN, and K-Means. The results in Table 9 show minimal differences (59.81–60.28), suggesting that TOKENTUNE is not sensitive to the choice of clustering algorithm and is robust across methods.

## I.5 EFFECT OF UPDATE CADENCE OF MAB

We study how the update cadence (*i.e.,* the number of samples selected before refreshing MAB scores) affects selection quality and downstream performance. We fix the total data selection budget to 50,000 examples and vary the number of samples selected per iteration. Table 17 summarizes the results.

Table 17: Sensitivity to the update cadence of MAB. We report the number of samples per iteration, total number of iterations to reach the fixed budget, and the average performance across benchmarks.

| Update Cadence (Sample Ratio / Iter.) | #Samples / Iter. | #Iterations | Avg. Performance |
|---|---|---|---|
| 0.5% | 250 | 200 | 60.12 |
| 1% | 500 | 100 | **60.28** |
| 2.5% | 1250 | 40 | 59.87 |
| 5% | 2500 | 20 | 59.45 |

We observe that updating UCB scores every 1% of the total budget yields the best average performance. When updates are too infrequent (e.g., every 2.5% or 5%), the accumulated reward estimates become stale, making the bandit over-exploit early high-reward clusters while neglecting newly emerging high-utility regions. Since UCB relies on the running average of observed rewards, delayed updates hinder its ability to adapt, ultimately degrading selection quality. On the other hand, overly frequent updates (e.g., every 0.5%) bring marginal gains at the cost of increased scheduling overhead. The 1% update cadence provides a favorable trade-off between reward estimation precision and exploration coverage. Furthermore, the performance remains stable across a reasonable range (0.5%–2.5%), demonstrating that our method is robust to this hyperparameter in a local neighborhood.

## J THE DESIGN DETAILS OF TOKENTUNE

### J.1 TOKENTUNE FRAMEWORK: CORE COMPONENTS

TOKENTUNE has three carefully designed core components.

❶ **Dual-Level Utility Function.** Considering that sample-level data selection methods overlook token heterogeneity while token-level ones fail to capture holistic sample value, we design a dual-level utility function that combines fine-grained token informativeness with principled sample selection. Specifically, it first leverages token-level indicators to capture learnable and uncertain tokens, and then constructs a sample-level utility by aggregating the token-level utilities over these learnable tokens, thereby avoiding a second-pass utility computation and reducing the overall annotation cost.

❷ **Adaptive Data Selection via MAB-Integrated Scheduler.** Evaluating the utility of each sample and token usually requires repeated model inference, which leads to prohibitive computational cost. To reduce this overhead and further scale efficiently to large datasets, we first partition the data pool into semantic clusters. Building on this, we introduce a multi-armed bandit scheduler that adaptively selects the most promising clusters, and only then applies dual-level utility estimation within each cluster to select the most informative samples.

❸ **Token-Aware Finetuning with Gated Optimization.** Not all tokens contribute to learning in the same way. Prior studies primarily focus on learnable tokens that provide strong supervision, but this narrow emphasis often leads to overfitting, as it ignores uncertain tokens-such as those admitting multiple correct answers. Training such ambiguous positions with standard cross-entropy forces the model to commit to a single label, thereby collapsing inherent diversity. To address this limitation, we propose a gated optimization strategy that differentiates token roles: learnable tokens are optimized with cross-entropy, ambiguous tokens are refined via self-distillation to preserve diversity, and uninformative tokens are suppressed to avoid noise amplification. For ambiguous tokens, self-distillation is particularly suitable because it optimizes the student to match a soft probability distribution from a teacher model, allowing probability mass to be spread over multiple plausible labels rather than collapsing it onto a single hard target as in standard cross-entropy.

## J.2 TOKEN-LEVEL UTILITY FUNCTION

**Token-level selection principles.** Our token-level utilities are designed to answer a simple question: *given a limited training budget, on which tokens does an additional update yield the largest marginal improvement?* Concretely, we follow two principles: (i) we would like to prioritize tokens whose further training is expected to produce a large reduction in loss *per token*, so that each gradient update is spent where it is most effective; (ii) among such tokens, we want to distinguish between those that are *under-learned but consistent* (single correct answer) and those that are *inherently multi-answer* (several plausible outputs), since the latter should not be forced into a single hard label.

**Definition J.1** (*Answer Uncertainty (AU)*). Some tokens are inherently ambiguous, admitting multiple plausible answers. To identify such positions, we model predictive uncertainty using an evidential Dirichlet distribution. For a token position $(i, j)$, let $\mathbf{z}(x_{i,j}) = (z_1(x_{i,j}), \ldots, z_K(x_{i,j}))$ denote the pre-softmax logits over the vocabulary. We map logits to non-negative evidence and obtain Dirichlet parameters $\boldsymbol{\alpha}(x_{i,j}) = (\alpha_1(x_{i,j}), \ldots, \alpha_K(x_{i,j}))$; specifically,

$$\alpha_k(x_{i,j}) = \max(0, z_k(x_{i,j})) + 1, \quad \alpha_0(x_{i,j}) = \sum_{k=1}^{K} \alpha_k(x_{i,j}). \tag{16}$$

We then define the answer uncertainty at $(i, j)$ as the *expected predictive entropy* of a categorical distribution $\mathbf{p}$ drawn from this Dirichlet:

$$AU(x_{i,j}) := \mathbb{E}_{\mathbf{p} \sim \text{Dir}(\boldsymbol{\alpha}(x_{i,j})+\mathbf{1})} \Big[ -\sum_{k=1}^{K} p_k \log p_k \Big], \tag{17}$$

where $\mathbf{1}$ is the all-ones vector. Using standard properties of the Dirichlet distribution, this expectation admits the closed-form expression

$$AU(x_{i,j}) = -\sum_{k=1}^{K} \frac{\alpha_k(x_{i,j})}{\alpha_0(x_{i,j})} \Big( \psi(\alpha_k(x_{i,j}) + 1) - \psi(\alpha_0(x_{i,j}) + 1) \Big), \tag{18}$$

where $\psi(\cdot)$ is the digamma function.

**Interpretation and comparison to entropy.** By construction, $\alpha_k(x_{i,j})$ can be interpreted as *evidence* supporting token $k$ at position $(i, j)$: large positive logits translate into large pseudo-counts, while low or negative logits contribute almost no evidence. The total evidence $\alpha_0(x_{i,j})$ encodes how confident the model is overall, whereas the relative magnitudes of $\alpha_k$ indicate whether this evidence is concentrated on one candidate or dispersed across several.

The quantity $AU(x_{i,j})$ is the expected entropy of a categorical distribution sampled from the Dirichlet with parameters $\boldsymbol{\alpha}(x_{i,j}) + \mathbf{1}$. It becomes large only when (i) the total evidence $\alpha_0(x_{i,j})$ is large (the model is confident), and (ii) this evidence is distributed over multiple candidates rather than concentrated on a single one. If the model is unsure and assigns low logits to all tokens, the evidence vector is small and $AU(x_{i,j})$ remains moderate despite the softmax distribution being nearly flat; if the model is confident and sharply focused on a single token, the expected entropy is small. Thus, high $AU(x_{i,j})$ specifically indicates *confident but multi-modal* beliefs, matching principle (ii) for inherently multi-answer positions.

This also explains why AU is preferable to (temperature-scaled) softmax entropy as an ambiguity indicator. Entropy depends only on normalized probabilities and cannot distinguish between high entropy due to lack of knowledge (low evidence spread over many tokens) and high entropy due to strong evidence for several distinct candidates. AU explicitly couples confidence (total evidence) and dispersion (how many tokens share it), allowing us to upweight truly multi-answer tokens while downweighting noisy, low-evidence ones. In our ablations, replacing AU with entropy consistently degrades performance and selects many low-evidence tokens as "ambiguous", supporting AU as a more faithful signal for token-aware training.

## J.3 SAMPLE-LEVEL UTILITY FUNCTION

A major drawback of existing model-aware data selection methods is that estimating sample utility requires repeated inference over the full dataset, leading to prohibitive computational latency at scale. To overcome this limitation, we build directly on the token-level feature $LG = \Delta \ell_{i,j}$ defined in Eq. 2 and construct a sample-level utility function without any additional inference.

**Token-level utility density.** For a sample $x_i = \{t_{i,1}, \ldots, t_{i,n_i}\}$, we first define a per-token notion of utility. Let

$$\Delta\ell_{i,j} := \ell_{\text{ref}}(x_{i,j}) - \ell_0(x_{i,j}), \tag{19}$$

be the reduction in loss of token $(i, j)$ when we move from the base model to the reference model, and let

$$b_{i,j} := \ell_0(x_{i,j}) = -\log p_{\theta_0}(y_{i,j} \mid x_i) \tag{20}$$

denote the baseline cross-entropy loss of that token under the base model $p_{\theta_0}$. Intuitively, $b_{i,j}$ measures how *difficult* this token is for the current model: tokens with large $b_{i,j}$ are ones to which the model assigns low probability (uncertain or often wrong), while tokens with small $b_{i,j}$ are already well mastered.

We then define the token-level utility density as

$$\rho_{i,j} := \frac{\Delta\ell_{i,j}}{b_{i,j}}. \tag{21}$$

This ratio performs a first normalization: it rescales the raw loss reduction $\Delta\ell_{i,j}$ by the baseline difficulty $b_{i,j}$ and can be interpreted as the *loss improvement per unit difficulty* of this token. In particular, $\rho_{i,j}$ is comparable across tokens with very different baseline losses, and it is invariant to any global rescaling of the loss (e.g., when switching between equivalent loss parameterizations).

**From tokens to samples.** Given any subset of token positions $S \subseteq \{1, \ldots, n_i\}$ within a sample, our goal at the sample level is to measure how much total loss reduction we obtain per unit of *total difficulty budget* in $S$. This leads to the following sample-level utility:

$$U_i(S) := \frac{\sum_{j \in S} \Delta\ell_{i,j}}{\sum_{j \in S} b_{i,j}}. \tag{22}$$

Thus $U_i(S)$ has the same semantics as the per-token density: it is the average loss improvement per unit of baseline difficulty in $S$, and is directly comparable across subsets with different lengths and difficulty profiles.

We can rewrite $U_i(S)$ as a weighted average of token densities:

$$U_i(S) = \sum_{j \in S} w_{i,j}(S)\, \rho_{i,j}, \quad w_{i,j}(S) := \frac{b_{i,j}}{\sum_{t \in S} b_{i,t}}. \tag{23}$$

Here $w_{i,j}(S)$ simply turns the baseline losses into a probability distribution over tokens in $S$. Equivalently,

$$U_i(S) = \mathbb{E}_{j \sim \pi_S}[\rho_{i,j}], \qquad \pi_S(j) = w_{i,j}(S). \tag{24}$$

Hence, $U_i(S)$ admits a clear semantic interpretation: it is the *expected loss improvement per unit of difficulty* when we pick a token from $S$ with probability proportional to its baseline loss. Tokens with larger baseline loss occupy a larger share of the total "difficulty budget" $\sum_{j \in S} b_{i,j}$ and therefore contribute proportionally more to the sample-level utility, which matches the intuition that harder tokens are both more costly and have more room for improvement.

**Uniqueness of the normalization.** Our construction can be seen as the unique sample-level extension of the token density that satisfies a small set of natural desiderata:

- **Consistency with token-level density.** When $S$ contains a single token, we require $U_i(\{j\}) = \rho_{i,j} = \Delta\ell_{i,j}/b_{i,j}$.
- **Budget-based aggregation.** The sample utility should depend on a subset $S$ only through the aggregate loss reduction $\sum_{j \in S} \Delta\ell_{i,j}$ and the aggregate difficulty $\sum_{j \in S} b_{i,j}$, reflecting the idea that we care about the total improvement given a total difficulty budget.

Under these conditions, any sample-level utility must have the form $U_i(S) = g\left(\frac{\sum_{j \in S} \Delta\ell_{i,j}}{\sum_{j \in S} b_{i,j}}\right)$ for some scalar function $g$. The consistency requirement $U_i(\{j\}) = \Delta\ell_{i,j}/b_{i,j}$ then forces $g(z) = z$, yielding exactly our definition of $U_i(S)$. Therefore, within this natural class of budget-based and scale-invariant utilities, our normalization is essentially unique.

Table 18: Summary of main notation used in TOKENTUNE.

| Symbol | Description |
|--------|-------------|
| $D$ | Full training data pool |
| $x_i$ | $i$-th input sequence (sample) |
| $n_i$ | Token length of sample $x_i$ |
| $x_{i,j}$ | $j$-th token in sample $x_i$ |
| $y_{i,j}$ | Target token at position $(i,j)$ |
| $\theta$ | Model parameters (generic) |
| $\theta_0$ | Current / base model parameters |
| $\theta_{\text{ref}}$ | Reference (teacher) model parameters |
| $\ell_\theta(x_{i,j})$ | Token-level loss under model $\theta$ |
| $\ell_0(x_{i,j})$ | Loss under current model $\theta_0$ |
| $\ell_{\text{ref}}(x_{i,j})$ | Loss under reference model $\theta_{\text{ref}}$ |
| $b_{i,j}$ | Baseline loss / difficulty: $b_{i,j} = \ell_0(x_{i,j})$ |
| $\Delta\ell_{i,j}$ | Learning gain: $\Delta\ell_{i,j} = \ell_0(x_{i,j}) - \ell_{\text{ref}}(x_{i,j})$ |
| $LG(x_{i,j})$ | Learning Gain at token $(i,j)$ (equal to $\Delta\ell_{i,j}$) |
| $\mathbf{z}(x_{i,j})$ | Logits vector at position $(i,j)$ |
| $z_k(x_{i,j})$ | Logit for vocabulary token $k$ at $(i,j)$ |
| $\alpha_k(x_{i,j})$ | Dirichlet evidence for token $k$ at $(i,j)$ |
| $\alpha_0(x_{i,j})$ | Total Dirichlet evidence: $\alpha_0 = \sum_k \alpha_k$ |
| $AU(x_{i,j})$ | Answer Uncertainty at token $(i,j)$ |
| $\rho_{i,j}$ | Token-level utility density: $\rho_{i,j} = \Delta\ell_{i,j}/b_{i,j}$ |
| $U_i(S)$ | Sample utility over token subset $S \subseteq \{1,\ldots,n_i\}$ |
| $U_k^{\text{Sample}}(x_i)$ | Sample utility using top-$k\%$ tokens by $\rho_{i,j}$ |
| $\hat{y}_{i,j}$ | Token label: 0 (uninformative), 1 (learnable), 2 (ambiguous) |
| $\tau_{LG}, \tau_{AU}$ | Thresholds for LG and AU in token labeling |
| $\{C_1,\ldots,C_K\}$ | Clusters of the data pool $D$ |
| $CS_i(t)$ | UCB cluster score of $C_i$ at iteration $t$ |
| $\bar{I}_i(t)$ | Average influence score of cluster $C_i$ at iteration $t$ |
| $T(C_i,t)$ | Number of times cluster $C_i$ is selected up to $t$ |
| $\gamma$ | Exploration coefficient in UCB scheduler |
| $S_i$ | Selected sample subset from cluster $C_i$ |
| $V$ | Vocabulary size |
| $T$ | Distillation temperature |
| $q(v)$ | Teacher distribution over vocabulary token $v$ |
| $\lambda$ | Trade-off between CE and distillation losses |

**Top-$k$ token subset.** In practice, not all tokens in a sample carry useful signal: very low-density tokens may correspond to noise or regions where the reference and base models already largely agree. To avoid dilution by such tokens, we focus on the subset of top-$k\%$ tokens ranked by $\rho_{i,j}$ within each sample. The final sample utility score used by our method is thus

$$U_k^{\text{Sample}}(x_i) := \frac{\sum_{j\in\text{top-}k(\rho)} \Delta\ell_{i,j}}{\sum_{j\in\text{top-}k(\rho)} b_{i,j}} = \frac{\sum_{j\in\text{top-}k(\rho)} \left(\ell_{\text{ref}}(x_{i,j}) - \ell_0(x_{i,j})\right)}{\sum_{j\in\text{top-}k(\rho)} \ell_0(x_{i,j})}. \tag{25}$$

This quantity can be interpreted as the *expected marginal improvement per unit difficulty budget* restricted to the most informative tokens of each sample, providing a stable and comparable notion of sample utility across examples with widely varying lengths and difficulty.

## K SUMMARY OF NOTATION

In this section, we summary the notation that TOKENTUNE used in Table 18.

