# OpenReview forum: "TokenTune: Dual-Level Utility Estimation for Scalable Data Selection in Instruction Tuning"
_ICLR.cc/2026/Conference — Submitted to ICLR 2026_

### Official Review · Reviewer_L7K1 · 2025-10-26

**Soundness:** 3
**Presentation:** 3
**Contribution:** 3
**Rating:** 6
**Confidence:** 4

**Summary:**

This paper addresses efficient data selection for instruction-tuning LLMs. The authors observe that existing selection methods either score at the sample level and ignore which tokens within each example are actually useful, or compute token-level utility but require full-dataset inference, which is too expensive at scale. To overcome this, they propose TokenTune, a framework using a dual-level utility score operating both at the token and sample levels: at the token level they identify “learnable tokens” and “multi-answer tokens” that still carry strong gradient signals and ensure diversity; at the sample level they derive a utility score from aggregated token signals, avoiding full-dataset inference. They also design a two-stage selection pipeline: clustering/information-prioritization followed by sample-level selection.

**Strengths:**

Introduces a thoughtful dual-level scoring concept, recognizing the importance of token-level signals in data utility.

The method addresses a real practical challenge: full-dataset scoring is expensive, so they pragmatically avoid it via aggregated token signals.

Demonstrates meaningful empirical gains (e.g., +3.8% on average using ~5% data) which is impressive in the instruction-tuning context.

The presentation is clear and the pipeline appears implementable for practitioners.

**Weaknesses:**

Token-level scoring may introduce selection bias: by focusing on “learnable tokens” and “multi-answer tokens”, rare but valuable tokens or edge cases might be under-selected. The paper currently provides limited analysis of diversity/coverage of selected subset.

Some hyperparameter choices (thresholds for token utility, clustering size, budget ratio) are not fully justified or ablated, which may impact reproducibility.

While selection uses ~5% of data, it remains unclear how this scales with pool sizes much larger than used in the experiments (e.g., millions of examples) and whether the method maintains its advantage.

**Questions:**

Could you provide a detailed breakdown of compute cost: (a) token-level scoring time, (b) clustering/selection time, (c) fine-tuning time for selected subset vs full dataset?

How robust is the token-level utility scoring to different model architectures or sizes? If the downstream fine-tune target model is much larger (or a different family) than the one used to estimate token signals, does performance degrade?

Have you analysed the selected subset in terms of diversity: e.g., task types, token vocabulary coverage, rare vs common tokens, difficulty levels? Could there be bias in selecting only “easy/high-gradient” tokens?

What is the sensitivity of results to budget ratio? If they selected 10% vs 2.5% vs 1% of the data, how does performance scale?

---

> ### Author Response · Authors · 2025-11-27
> **Response to Reviewer L7K1 (1/5)**
>
> > **Q1: Could you provide a detailed breakdown of compute cost: (a) token-level scoring time, (b) clustering/selection time, (c) fine-tuning time for selected subset vs full dataset?**
>
> **Response**
>
> Thank you for the helpful question. We would like to clarify that our original submission already reports both the total selection and fine-tuning time of TokenTune and several existing baselines. These results are shown in **Figure 5** of paper a nd demonstrate that TokenTune achieves better performance than other methods with significantly lower computational cost.
>
> To further address the reviewer’s concern, we provide a more detailed breakdown of the selection time into two parts: (1) clustering and (2) token-level scoring. The results are summarized in the table below.
>
> * Our method requires only 1.7 hours for clustering and 13.1 hours for scoring, totaling 14.8 hours for selection, which is significantly faster than all prior methods (e.g., IFD: 196.7h, ZIP: 111h, Entropy: 129.8h).
> * Combined with 3.6h of fine-tuning, the total cost of our approach is only 18.4 hours, which is over **8× faster than training on the full dataset (152h)** while achieving better performance (60.3 vs. 59.7).
>
>
> |                          | **Component**       | SelectIT |  ZIP   | Entropy |  IFD   | Deita | **Ours** | Full Dataset |
> |:------------------------:|:-------------------:|:--------:|:-----:|:-------:|:------:|:-----:|:--------:|:-------------:|
> | **Latency (h)**          | Clustering (h)      |    -     |   -   |    -    |  16.3  |   -   |   1.7    |       -       |
> |                          | Scoring (h)         |   165    |  111  |  129.8  | 196.7  | 17.0  |   13.1   |       -       |
> |                          | Finetuning (h)      |   3.7    |  3.6  |   3.4   |  3.9   |  4.1  |   3.6    |      152      |
> |                          | **Total Time (h)**  |  165.4   | 114.6 |  133.2  | 216.9  | 21.1  | **18.4** |      152      |
> | **Avg. Performance (%)** |           -          |    -     | 56.8  |  56.4   |  55.1  | 57.5  | **60.3** |     59.7      |
>
>
> ---
>
> > **W2: Some hyperparameter choices (thresholds for token utility, clustering size, budget ratio) are not fully justified or ablated, which may impact reproducibility.**
>
> We thank the reviewer for the thoughtful suggestions. We have revised the paper accordingly and added the requested details in **Appendix C**  and **Appendix I**​.
>
> We thank the reviewer for pointing out the importance of hyperparameter transparency and reproducibility. In the revised manuscript, we have provided detailed reporting of all default values and search ranges for key hyperparameters in **Appendix C**​, including the thresholds for token utility, clustering size, and budget ratio.
>
> To further assess the robustness of our method, we conducted a comprehensive sensitivity analysis for these hyperparameters, as presented in **Appendix I**​. Below, we report the sensitivity results on token utility, clustering size, and budget ratio. The results show that performance remains generally stable within the local neighborhood of each hyperparameter, demonstrating that our method is robust to reasonable variations and does not rely on precise tuning.
>
> Please refer to **Appendix I** for complete results and discussion.
>
>
>
> | Threshold (Token Utility) | TyDiQA | HellaSwag | ARC-C | BoolQ | GSM8K | HumanEval | LogiQA | Average |
> |:-------------------------:|:------:|:---------:|:-----:|:-----:|:-----:|:---------:|:------:|:-------:|
> | 0.7                       | 57.78  | 61.56     | 52.54 | 84.00 | 58.04 |   74.54   | 26.82  | 59.33   |
> | 0.6                       | 57.16  | 61.55     | 53.92 | 84.40 | 60.49 |   76.09   | 28.37  | **60.28** |
> | 0.5                       | 57.50  | 61.69     | 53.57 | 84.03 | 57.81 |   75.93   | 27.13  | 59.67   |
> | 0.4                       | 55.88  | 61.45     | 52.89 | 84.37 | 57.50 |   73.52   | 24.65  | 58.61   |
> | 0.3                       | 54.17  | 60.02     | 51.38 | 82.47 | 55.73 |   74.62   | 25.78  | 57.74   |
>
>
> | Sample Ratio | TyDiQA | HellaSwag | ARC-C | BoolQ | GSM8K | HumanEval | LogiQA | Average |
> |:--------------:|:------:|:---------:|:-----:|:-----:|:-----:|:---------:|:------:|:-------:|
> | 1%             | 51.66  | 61.56     | 54.61 | 84.43 | 54.13 |   74.72   | 27.29  | 58.34   |
> | 2.5%           | 55.78  | 61.03     | 53.74 | 83.94 | 56.28 |   75.13   | 28.01  | 59.13   |
> | 5%             | 57.16  | 61.55     | 53.92 | 84.40 | 60.49 |   76.09   | 28.37  | **60.28** |
> | 7%             | 54.10  | 61.64     | 54.44 | 85.89 | 57.89 |   74.36   | 27.60  | 59.42   |
> | 10%            | 59.15  | 61.13     | 53.40 | 83.89 | 57.20 |   76.04   | 26.82  | 59.66   |
>
>
> | Different Clustering Algorithm | Kmeans | DBSCAN | GMM   |
> |:-----------------------------:|:------:|:------:|:-----:|
> | Avg. Performance              | 58.73  | 58.91  | **60.28** |

---

> ### Author Response · Authors · 2025-11-27
> **Response to Reviewer L7K1 (2/5)**
>
> > **Q3: What is the sensitivity of results to budget ratio? If they selected 10% vs 2.5% vs 1% of the data, how does performance scale?**
>
> **Response**
>
> Thank you for the insightful question. Following your suggestion, we conducted a comprehensive evaluation to analyze the sensitivity of TokenTune under different budget constraints. Specifically, we compared performance at 1%, 2.5%, 5% and 10% data selection ratios, and benchmarked TokenTune against several strong baselines under each setting.
>
> As shown below, TokenTune consistently outperforms other baselines across all budget levels. Notably, the performance gap between TokenTune and other baselines is more pronounced at lower budgets (e.g., 1% and 2.5%), demonstrating its effectiveness in prioritizing high-utility data under resource-limited scenarios. At higher budgets (e.g., 10%), all methods improve, but TokenTune still achieves the best performance overall.
>
> These results confirm that TokenTune is not only scalable across different data budgets, but also particularly advantageous when training resources are limited.
>
> We have added this analysis and the corresponding comparison table in the revised manuscript. Please refer to **Appendix G.1** for more details.
>
> | Sample Budget Ratio | Method | TyDiQA | HellaSwag | ARC-C | BoolQ | GSM8K | HumanEval | LogiQA | Average |
> |:-------------------:|:------:|:------:|:---------:|:-----:|:-----:|:-----:|:---------:|:------:|:-------:|
> | 1%   | IFD    | 39.79 | 60.38 | 51.59 | 81.69 | 51.53 | 72.34 | 27.13 | 54.92 |
> |      | Instag | 42.37 | 60.27 | 51.77 | 83.17 | 54.59 | 75.23 | 26.82 | 56.32 |
> |      | Deita  | 46.56 | 60.65 | 52.37 | 83.20 | 56.66 | 72.19 | 29.61 | 57.32 |
> |      | **TokenTune (Ours)**   | 51.66 | 61.56 | 54.61 | 84.43 | 56.13 | 74.72 | 27.29 | **58.34** |
> | 2.5% | IFD    | 40.93 | 60.47 | 51.59 | 82.09 | 50.38 | 71.73 | 26.82 | 54.86 |
> |      | Instag | 44.78 | 60.22 | 51.68 | 83.20 | 57.66 | 73.20 | 28.22 | 56.99 |
> |      | Deita  | 45.60 | 61.00 | 51.08 | 82.12 | 58.26 | 72.60 | 28.68 | 57.05 |
> |      | **TokenTune (Ours)**   | 55.78 | 61.03 | 53.74 | 83.94 | 56.28 | 75.13 | 28.01 | **59.13** |
> | 5%   | IFD    | 38.55 | 60.35 | 49.87 | 82.21 | 57.04 | 71.60 | 26.36 | 55.14 |
> |      | Instag | 44.97 | 60.66 | 50.39 | 84.34 | 58.80 | 75.50 | 26.67 | 57.33 |
> |      | Deita  | 44.81 | 60.74 | 52.11 | 82.86 | 57.35 | 74.60 | 30.08 | 57.51 |
> |      | **TokenTune (Ours)**   | 57.16 | 61.55 | 53.92 | 84.40 | 60.49 | 76.09 | 28.37 | **60.28** |
> | 7%   | IFD    | 38.48 | 60.20 | 49.35 | 82.09 | 55.28 | 72.30 | 25.27 | 54.71 |
> |      | Instag | 43.04 | 60.53 | 51.51 | 83.94 | 58.96 | 75.35 | 26.67 | 57.14 |
> |      | Deita  | 42.10 | 60.70 | 51.42 | 83.54 | 60.03 | 71.41 | 29.61 | 56.97 |
> |      | **TokenTune (Ours)**  | 54.10 | 61.64 | 54.41 | 85.89 | 57.89 | 74.36 | 27.60 | **59.42** |
> | 10%  | IFD    | 39.48 | 60.18 | 50.73 | 82.15 | 55.44 | 73.41 | 26.82 | 55.46 |
> |      | Instag | 44.17 | 60.02 | 52.54 | 83.17 | 58.26 | 74.68 | 26.08 | 56.61 |
> |      | Deita  | 44.57 | 60.80 | 51.94 | 83.29 | 59.80 | 72.46 | 28.99 | 57.41 |
> |      | **TokenTune (Ours)**  | 59.15 | 61.13 | 53.40 | 83.89 | 57.20 | 76.04 | 26.82 | **59.66** |

---

> ### Author Response · Authors · 2025-11-27
> **Response to Reviewer L7K1 (3/5)**
>
> > **W1：Token-level scoring may introduce selection bias: by focusing on “learnable tokens” and “multi-answer tokens”, rare but valuable tokens or edge cases might be under-selected. The paper currently provides limited analysis of diversity/coverage of selected subset.**
>
> **Response**
>
> We thank the reviewer for raising this important concern. We agree that maintaining diversity in the selected subset is crucial, and we have taken multiple steps in our design to mitigate selection bias and ensure broad coverage.
>
> * **Cluster-level Diversity via MAB Sampling:** At the sample level, we first perform clustering over all samples and apply a multi-armed bandit (MAB) strategy to select promising clusters for utility estimation. The MAB algorithm naturally balances cluster utility and sampling frequency, which prevents over-selection of high-utility clusters and promotes diversity across clusters. Since our dataset spans multiple domains, this strategy encourages the selection of samples from a wide variety of task types.
> * **Token-level Diversity from Task-specific Learning Signals:** Our Learning Gain (LG) metric identifies tokens with the highest estimated learning gain within each token. Intuitively, these tokens are often task-specific and informative, helping the model quickly adapt to different task types. Because MAB sampling already ensures coverage across diverse tasks, the LG-based token selection further reinforces diversity at the token level.
> * **Answer Uncertainty (AU) Promotes Semantic Diversity:** The AU metric identifies ambiguous tokens where multiple plausible answers may exist. These are precisely the tokens where standard training (e.g., cross-entropy loss) may overfit to a single observed label. By detecting and treating these tokens differently using self-distillation, AU helps the model preserve semantic diversity in its output distribution. This design choice explicitly addresses the risk of collapsing onto deterministic outputs .
>
>
> ---
>
> > **Q3: Have you analysed the selected subset in terms of diversity: e.g., task types, token vocabulary coverage, rare vs common tokens, difficulty levels? Could there be bias in selecting only “easy/high-gradient” tokens?**
>
> **Response**
>
> We thank the reviewer for this important question. As discussed in our response to W1, our method incorporates several design choices to ensure diversity at both the sample and token levels. Here, we would like to clarify that our selection strategy is not biased toward easy tokens. Instead, it is designed to prioritize tokens that offer the greatest potential improvement in model performance. This naturally includes tokens of varying difficulty levels.
>
> To illustrate this, consider two hypothetical tokens with the same utility score of 0.1. One token has a loss reduction of $l_0 - l_{ref} = 0.2 - 0.1$, while the other has $l_0 - l_{ref} = 4.2 - 4.1$. Despite the difference in absolute difficulty, both yield equal learning gain according to our scoring function.

---

> ### Author Response · Authors · 2025-11-27
> **Response to Reviewer L7K1 (4/5)**
>
> > **Q2: How robust is the token-level utility scoring to different model architectures or sizes? If the downstream fine-tune target model is much larger (or a different family) than the one used to estimate token signals, does performance degrade?**
>
> **Response**
>
> We thank the reviewer for raising this important point. To assess the robustness of our token-level utility scoring under architecture and scale mismatches, we conducted additional experiments under two settings:
>
> * ​**Cross-Architecture Setting**​: We compute token-level utilities using LLaMA-3.1-8B and fine-tune with Qwen2-7B.
> * ​**Cross-Scale Setting**​: Following Superfilter[1], we select data using GPT2 (a much smaller model) and fine-tune with LLaMA-3.1-8B.
>
> As shown in the results below, our method consistently outperforms baseline selection strategies across both settings. These results suggest that although our design is optimized for model-aware scoring, the learned signals are robust enough to transfer across different model sizes and families.
>
> We include full results in **Table 12** and **Table 13** of the revised manuscript and discuss in​ **Appendix H**​.
>
>
> | Selection Method (Backbone: LLaMA3.1-8B) | TyDiQA | HellaSwag | ARC-C | BoolQ | GSM8K | humaneval | logiqa | Average |
> |:---------------------------------------:|:------:|:---------:|:-----:|:-----:|:-----:|:---------:|:------:|:-------:|
> | IFD                                     | 47.88  | 58.96     | 47.03 | 83.91 | 78.56 | 76.43     | 30.23  | 60.43   |
> | SelectIT                                | 50.98  | 58.00     | 49.78 | 83.43 | 76.79 | 77.49     | 29.15  | 60.80   |
> | Entropy                                 | 46.19  | 58.48     | 45.56 | 85.27 | 76.92 | 76.96     | 28.99  | 59.77   |
> | TL                                      | 50.66  | 59.07     | 48.75 | 84.47 | 77.72 | 75.26     | 32.71  | 61.23   |
> | **TokenTune (Ours)**        | 49.31  | 59.80     | 52.89 | 84.50 | 78.42 | 78.24     | 30.70  | **61.98** |
>
> | Selection Method (Backbone: GPT2) | TyDiQA | HellaSwag | ARC-C | BoolQ | GSM8K | humaneval | logiqa | Average |
> |:---------------------------------:|:------:|:---------:|:-----:|:-----:|:-----:|:---------:|:------:|:-------:|
> | IFD                               | 43.57  | 60.49     | 52.89 | 83.26 | 55.28 | 74.07     | 26.06  | 56.52   |
> | SelectIT                          | 42.73  | 60.04     | 53.18 | 83.14 | 52.32 | 73.92     | 26.35  | 55.95   |
> | Entropy                           | 43.51  | 60.37     | 51.94 | 84.47 | 51.15 | 71.52     | 25.89  | 55.55   |
> | TL                                | 47.48  | 60.51     | 50.34 | 82.49 | 51.32 | 73.43     | 26.74  | 56.04   |
> | **TokenTune (Ours)**   | 55.83  | 61.31     | 53.14 | 83.97 | 57.81 | 75.89     | 27.99  | **59.42** |

---

> ### Author Response · Authors · 2025-11-27
> **Response to Reviewer L7K1 (5/5)**
>
> > **W3: While selection uses \~5% of data, it remains unclear how this scales with pool sizes much larger than used in the experiments (e.g., millions of examples) and whether the method maintains its advantage.**
>
> **Response**
>
> We thank the reviewer for raising this important point. We would like to clarify that our original submission already includes results on two datasets, one of which, **OpenHermes**, contains over one million training examples. This setup satisfies the large-scale pool condition suggested by the reviewer. In this setting, our method achieves the highest average performance (65.51%) among all methods, demonstrating its effectiveness even under large-scale supervision.
>
> To further address the concern, we conducted an additional scalability analysis by varying the pool size on Tulu3. Specifically, we tested with 200K, 500K, 700K, and approximately 1M examples. The results in **Figure 8** in our revised paper show that TokenTune consistently outperforms baseline methods across all pool sizes.
>
> We have included these discussions in the revised version (​**Appendix D**​).
>
> | Method             | TyDiQA | HellaSwag | ARC-C | BoolQ | GSM8K | HumanEval | LogiQA | Avg.  |
> |:------------------:|:------:|:---------:|:-----:|:-----:|:-----:|:---------:|:------:|:-----:|
> | Base               | 22.80  | 59.92     | 50.82 | 82.18 | 50.31 | 69.28     | 26.51  | 51.69 |
> | Random             | 49.44  | 59.31     | 51.75 | 82.16 | 58.27 | 71.98     | 27.46  | 62.15 |
> | IFD                | 41.97  | 60.21     | 52.45 | 82.89 | 52.45 | 69.80     | 27.75  | 55.36 |
> | ZIP                | 47.82  | 60.33     | 53.32 | 84.84 | 52.76 | 71.39     | 27.91  | 56.91 |
> | Entropy            | 51.45  | 60.41     | 50.04 | 83.17 | 56.74 | 72.49     | 26.05  | 57.19 |
> | Instag             | 47.21  | 60.31     | 52.28 | 83.19 | 60.76 | 72.50     | 26.82  | 62.71 |
> | TL                 | 46.61  | 60.89     | 52.71 | 83.14 | 54.36 | 73.36     | 28.53  | 57.09 |
> | SelectIT           | 49.15  | 60.33     | 52.49 | 83.94 | 60.03 | 70.79     | 27.91  | 57.81 |
> | Deita              | 42.50  | 60.80     | 51.85 | 83.17 | 54.59 | 72.92     | 29.77  | 60.97 |
> | **TokenTune (Ours)** | **53.56** | **61.49** | **52.63** | **85.05** | **65.62** | **74.69** | 28.37 | **65.51** |

---

### Official Review · Reviewer_BxaP · 2025-10-28

**Soundness:** 2
**Presentation:** 3
**Contribution:** 3
**Rating:** 4
**Confidence:** 2

**Summary:**

This paper presents TokenTune, a method to select informative data samples and prune out uninformative tokens from those selected data samples. Their solution is a hierarchical pruning method that selects samples and tokens based on the learning gain (the amount of information to learn from it, measured by the loss between the current and reference models) and the answer uncertainty (the ambiguity of a token which could lead the model to different answers, measured by an evidential Dirichlet distribution over the logits) -- both of which are metrics that they propose. They use a multi-armed bandit to select data: the data is clustered, the bandit learns to select a cluster based on a cluster score, and then samples are selected based on the token-level utility and the sample-level utility. Their experiments show that TokenTune is model agnostic, efficient, and effective.

**Strengths:**

- The problem is interesting
- The dual nature of the solution is creative
- The experimentation covers many datasets and baselines.
- The takeaways are well organized

**Weaknesses:**

- No error bars reported in the tables
- The LLMs chosen are all the same size, not sure how the performance would vary with smaller and larger models
- I might be missing something but I could not find the experimentation details for the MAB (hyperparameters)

**Questions:**

1. Is there any literature that supports the definition of Answer Uncertainty (equation 2). Why does the digamma function help measure the ambiguity of tokens?
2. I might be mistaken, but if you prune out unimportant tokens for fine-tuning, does that effect the ability of the LLM to generate grammatically correct English? Does your current evaluation reflect that?
3. Could you elaborate on the influence score $I_i(t)$? What does it mean, how is it defined, and how is it initialized?

---

> ### Author Response · Authors · 2025-11-27
> **Response to Reviewer BxaP (1/4)**
>
> > **W2: The LLMs chosen are all the same size, not sure how the performance would vary with smaller and larger models**
>
> **Response**
>
> We thank the reviewer for raising this point. To address this concern, we have added experiments on both a **larger model (LLaMA-2-13B)** and a ​**smaller model (LLaMA-3.2-3B)**.
>
> As shown  below, TokenTune consistently improves performance across scales: +1.09 on the 13B model and +2.33 on the 3B model compared to the strongest baselines under the same budget. These results indicate that our method generalizes well regardless of model capacity, and the performance trends observed in the main paper remain stable across smaller and larger LLMs.
>
> Table I. Comparison of performance on LLaMA-3.2-3
> | Method   | TyDiQA | HellaSwag | ARC-C | BoolQ | GSM8K | HumanEval | LogiQA | Average |
> |:--------:|:------:|:---------:|:-----:|:-----:|:-----:|:---------:|:------:|:-------:|
> | **Model: LLaMA-3.2-3B** |||||||||
> | Base     | 32.67  | 55.10     | 42.20 | 73.01 | 28.48 |   50.11   | 22.17  | 43.39   |
> | Random   | 38.96  | 55.21     | 41.95 | 73.68 | 29.01 |   57.13   | 24.51  | 45.78   |
> | DS2      | 39.96  | 55.31     | 44.27 | 74.86 | 29.94 |   55.71   | 23.72  | 46.25   |
> | Deita    | 31.07  | 55.74     | 42.89 | 75.60 | 32.47 |   56.90   | 24.34  | 45.57   |
> | ZIP      | 41.60  | 55.05     | 41.69 | 75.79 | 28.02 |   53.08   | 23.26  | 45.50   |
> | Entropy  | 40.01  | 54.37     | 41.12 | 74.36 | 28.62 |   55.62   | 23.91  | 45.43   |
> | Instag   | 37.02  | 55.40     | 42.46 | 72.85 | 30.76 |   56.17   | 24.50  | 45.59   |
> | CaR      | 38.87  | 54.99     | 41.95 | 76.28 | 30.93 |   56.63   | 25.89  | 46.51   |
> | TL       | 39.87  | 54.73     | 42.62 | 74.09 | 29.01 |   57.01   | 23.98  | 45.90   |
> | **TokenTune (Ours)**     | 49.13  | 55.55     | 43.58 | 77.95 | 32.39 |   59.06   | 24.19  | **48.84** |
>
> Table II. Comparison of performance on LLaMA-2-13B
> | Method   | TyDiQA | HellaSwag | ARC-C | BoolQ | GSM8K | HumanEval | LogiQA | Average |
> |:--------:|:------:|:---------:|:-----:|:-----:|:-----:|:---------:|:------:|:-------:|
> | **Model: LLaMA-2-13B** |||||||||
> | Base     | 31.20  | 60.01     | 47.72 | 80.91 | 24.12 |   31.22   | 25.89  | 43.01   |
> | Random   | 35.96  | 60.29     | 48.75 | 82.09 | 32.01 |   39.61   | 28.68  | 46.77   |
> | DS2      | 36.11  | 60.36     | 50.90 | 81.93 | 21.36 |   38.37   | 25.43  | 44.92   |
> | Deita    | 34.86  | 60.95     | 49.01 | 82.21 | 28.87 |   40.88   | 26.82  | 46.23   |
> | ZIP      | 40.08  | 60.21     | 50.65 | 82.80 | 26.19 |   36.49   | 27.13  | 46.22   |
> | Entropy  | 42.37  | 60.63     | 48.14 | 81.99 | 27.48 |   37.82   | 26.93  | 46.48   |
> | Instag   | 37.03  | 60.46     | 48.92 | 82.06 | 28.43 |   39.04   | 27.91  | 46.26   |
> | CaR      | 37.55  | 60.22     | 47.98 | 82.06 | 30.55 |   44.57   | 31.63  | 47.79   |
> | TL       | 41.26  | 60.47     | 49.53 | 82.37 | 29.82 |   34.16   | 28.06  | 46.52   |
> | **TokenTune (Ours)**    | 44.86  | 60.79     | 50.71 | 82.01 | 31.73 |   44.02   | 28.07  | **48.88** |
>
>
>  >**W1:No error bars reported in the tables**
>
> **Response**
>
> Thank you for the suggestion. In our experiments, we would like to clarify that we ran each model with three different random seeds and reported the average performance in all tables.

---

> ### Author Response · Authors · 2025-11-27
> **Response to Reviewer BxaP (2/4)**
>
> > **Q3: Could you elaborate on the influence score $I_i(t)$? What does it mean, how is it defined, and how is it initialized?**
>
> **Response**
>
> We thank the reviewer for the question. Below we provide a structured explanation of what the influence score represents, how it is defined, and how it is initialized.
>
> **1. What does $I_i(t)$ mean?**
>
> **1.1 Preliminary: MAB and Its Key Idea**
>
> Our scheduler is based on the **Multi-Armed Bandit (MAB)** framework, which is an effective framework that makes decisions over time under*uncertainty. This consists of \( N \) possible actions, each known as an arm.
>
> Each arm is assigned a score consisting of two components:
> - an **exploitation** term, which reflects the estimated reward of selecting the arm
> - an **exploration** term, which encourages selecting under-visited arms to maintain diversity
>
> The key idea of MAB is to balance exploration and exploitation to maximize their overall reward $\mathrm{UCB}_i(t)$ throughout the given time span.
>
> The full score is computed as:
> $$
> \mathrm{UCB}_i(t) = I_i(t) + \gamma \cdot B_i(t)
> $$
> where:
> - $I_i(t)$: the **exploitation term**, representing the estimated reward (influence score) of selecting cluster $C_i$ at time $t$
> - $B_i(t)$: the **exploration term**, encouraging selection of clusters that have been sampled less frequently
>
>
> **1.2 Motivation for using MAB in our setting**
>
> A straightforward approach to data selection is to compute utility scores for all samples and then select the top-performing ones. However, in large-scale instruction tuning, this becomes computationally infeasible. As shown in the table below, when the dataset reaches the million scale, full-dataset utility estimation can require more time and resources than directly fine-tuning on the entire dataset.
>
> To address this limitation, we incorporate the **Multi-Armed Bandit (MAB)** framework into our selection process. Rather than evaluating every sample, we let MAB guide the search toward promising samples of the dataset. The exploitation term prioritizes samples that have yielded high utility in previous iterations, while the exploration term ensures coverage by promoting under-visited clusters. This mechanism allows TokenTune to reduce computation significantly, without sacrificing performance or diversity in the selected data.
>
> |                      |  IFD  | SelectIT |  ZIP  | Entropy | Full Data (939k) |
> |:--------------------:|:-----:|:--------:|:-----:|:-------:|:----------------:|
> | **Selection Time (h)** | 213   |   165    | 111   |  129    |        0         |
> | **Training Time (h)**  | 3.9   |   3.7    | 3.6   |  3.4    |       152        |
>
>
>
>
>
> **1.3 Bridging MAB with data selection**
>
> To naturally connect data selection and MAB, we first cluster all samples in the candidate dataset. Thus, each cluster is rewarded as an individual arm in the bandit setting. At each iteration, the scheduler selects one promising cluster to explore or exploit, based on its UCB score.
>
> To quantify the reward associated with a cluster, we define an **influence score** $I_i(t)$, which measures how informative cluster $C_i$ has been up to iteration \( t \). At each iteration, a small subset of samples is drawn from a selected cluster, their utilities are computed, and their average is used to update $I_i(t)$. This score serves as the exploitation signal in the UCB rule.
>
> As a result, the MAB scheduler gradually shifts computational focus toward informative regions of the data pool, while still preserving diversity through exploration.
>
>
> **2. How is $I_i(t)$ Defined?**
>
> For each cluster \( C_i \), we define its influence score \( I_i(t) \) as the **running average of the sample-level utility scores** for all samples drawn from that cluster \( C_i \) up to iteration \( t \). Formally,
>
> $$
> I_i(t) = \frac{1}{T(C_i, t)} \sum_{s=1}^{t} \sum_{x \in S_i(s)} U_\text{Sample}(x),
> $$
> where:
> - $S_i(s) \subset C_i$ denotes the set of samples drawn from cluster $C_i$ at iteration $s$
> - $T(C_i, t) = \sum_{s=1}^{t} |S_i(s)|$ is the total number of samples drawn from $C_i$ up to iteration $t$
> - $U_{\text{Sample}}(x)$ is the sample-level utility defined in Equation (5)
>
> **3. How is $I_i(t)$ Initialized?**
>
> At the beginning of the selection process, before any samples have been drawn, we initialize the influence score of all clusters uniformly as:
> $$
> I_i(0) = 0, \quad T(C_i, 0) = 0 \quad \text{for all } i.
> $$
> This initialization ensures that all clusters start with equal priority. Since the exploitation term $I_i(t)$ is initially zero, the exploration term in the UCB score dominates in the early rounds. This encourages the scheduler to explore a broad range of clusters at the beginning, enabling a diverse estimate of utility across the dataset. As more data is collected and $T(C_i, t)$ increases, the influence score becomes a progressively refined and stable estimate of the cluster’s actual average utility.

---

> ### Author Response · Authors · 2025-11-27
> **Response to Reviewer BxaP (3/4)**
>
> > **W3: I might be missing something but I could not find the experimentation details for the MAB (hyperparameters)**
>
> **Response**
>
> We thank the reviewer for pointing this out and apologize for the omission in the initial version. In the revised manuscript, we have added the missing experimentation details for the MAB module, including the values and roles of key hyperparameters.
>
> In addition, we have added a sensitivity analysis on two important parameters of MAB: the number of MAB arms and the exploration rate.
>
> * **Effect of the Number of MAB Arms K.** In our setup, each arm corresponds to a cluster. We use the Elbow method to guide the choice of arms/clusters k, which eliminates the need for manual adjustment. The results below show that too few clusters (e.g., k=100) lead to high variance and under-representation of data, while too many clusters (e.g., k=5000) introduce redundancy and reduce exploration efficiency. A moderate choice of k=1000 provides the best balance between selection diversity and computational efficiency.
> * **Effect of Exploration Rate 𝛾 of MAB.**  Our approach employs 𝛾  to balance the diversity and quality during cluster sampling. The results below show that when 𝛾  is small, the MAB framework prioritizes high-influence clusters and risks local optima due to reduced diversity. Conversely, when 𝛾 is large, it overemphasizes diversity at the expense of quality, limiting model performance gains.
>
> We have included the above discussion into **Appendix I** of the revised paper.
>
> Table I. Effect of MAB Arms / Clustering Size.
> | MAB Arms K | 100   | 500   | 1000  | 2000  | 3000  | 5000  |
> |:----------------------------:|:-----:|:-----:|:-----:|:-----:|:-----:|:-----:|
> | Performance (Avg.)          | 58.73 | 58.91 | **60.28** | 60.01 | 59.67 | 59.04 |
>
>
> Table II. Effect of Exploration Rate of MAB.
> | Exploration Rate γ of MAB | 0.0005 | 0.001  | 0.005  | 0.01   | 0.05   |
> |:--------------------------:|:------:|:------:|:------:|:------:|:------:|
> | Avg. Performance           | 59.02  | **60.28** | 59.67  | 59.04  | 57.88  |
>
>
> > **Q2: I might be mistaken, but if you prune out unimportant tokens for fine-tuning, does that effect the ability of the LLM to generate grammatically correct English? Does your current evaluation reflect that?**
>
> **Response**
>
> **1. Our Approach vs Direct Token Pruning**
>
> We thank the reviewer for raising this point. We agree with the reviewer’s concern. In fact, many existing token-pruning works do remove tokens at the data level, and this often leads to the problem the reviewer highlighted: the natural structure of sentences becomes disrupted, which can negatively affect grammatical fluency. Recognizing this limitation, we deliberately designed TokenTune to avoid data-level token removal and instead adopt a different strategy that preserves complete input sequences while still reducing the influence of uninformative tokens during training.
>
> Specifically, TokenTune does not prune tokens at the data or input level. Instead, during finetuneing, all tokens are fed into the model in the forward pass, while only uninformative tokens are **masked in the backward pass**, ensuring they do not contribute gradients or affect parameter updates.
>
> **2. Why TokenTune Does Not Affect Grammatical Quality?**
>
> To the best of our knowledge, the ability of an LLM to generate grammatically correct text is primarily learned during the forward pass of supervised fine-tuning. In this phase, the model conditions on the full prefix and predicts the next token. TokenTune keeps this process unchanged: the model processes all tokens in the forward pass, so the linguistic signals necessary for grammatical fluency remain intact. The masking applies only in the backward pass, where gradients from uninformative tokens are suppressed. This preserves grammatical correctness by retaining complete input context, while improving training efficiency by reducing noisy gradient updates.
>
> **3. Additional Experiment**
>
> To further validate this point, we have conducted an additional experiment comparing two settings:
>
> (a) removing unimportant tokens before supervised fine-tuning;
>
> (b) applying our gradient-masking strategy while retaining full input sequences.
>
> The results show that data-level token removal leads to noticeable degradation in fluency-related metrics and overall performance, whereas TokenTune maintains grammatical quality and achieves higher accuracy across benchmarks.
>
> | Method           | TydiQA | HellaSwag | ARC-C | BoolQ | GSM8K | humaneval | logiqa | Average |
> |------------------|--------|-----------|--------|--------|--------|------------|--------|---------|
> | Token Pruning    | 53.48  | 60.38     | 52.09  | 82.66  | 55.35  | 74.17      | 25.12  | 57.61   |
> | **Gradient-Masking** | 57.16  | 61.55     | 53.92  | 84.40  | 60.49  | 76.09      | 28.37  | **60.28** |

---

> ### Author Response · Authors · 2025-11-27
> **Response to Reviewer BxaP (4/4)**
>
> > **Q1: Is there any literature that supports the definition of Answer Uncertainty (equation 2). Why does the digamma function help measure the ambiguity of tokens?**
>
> **Response**
>
> We thank the reviewer for the insightful question regarding the definition of Answer Uncertainty (AU) and the use of the digamma function in Equation (2). We have also included this theory in **Section 3.1**.
>
> Below, we clarify the motivation, theoretical support, and design rationale behind this component.
>
> **1. Literature Basis for the AU Metric**
>
> The design of our AU metric is inspired by the evidential uncertainty framework introduced in [1]. This framework models the predictive distribution over tokens as a Dirichlet distribution, where model logits are interpreted as non-negative evidence values.
>
> **2. Motivation for Using AU Metric in Our Setting**
>
> Building on this idea, we adapt the expected entropy of the Dirichlet distribution to construct a token-level ambiguity score, with the goal of identifying multi-answer tokens—positions where the model assigns high confidence to multiple plausible next-token candidates. Detecting such tokens is crucial: if standard cross-entropy loss is applied to them without adjustment, the model will overfit the observed answer and penalize equally valid alternatives. This may lead to reduced generalization and loss of output diversity during inference.
>
> **3. Why does the digamma function help measure the ambiguity of tokens?**
>
> We would like to clarify that it is not the digamma function itself that directly measures token ambiguity. Instead, the digamma function naturally arises in the closed-form expression of the expected entropy when modeling the predictive distribution as a Dirichlet distribution.
>
>
> **3.1 Why the Digamma Function Appears in Our Formulation**
>
> For a token position $(i, j)$ with target token $x_{i,j}$, let $\mathbf{z}(x_{i,j}) = \left(z_1(x_{i,j}), \dots, z_K(x_{i,j})\right)$ be the logits and map them to non-negative evidence:
>
> $$
> \alpha_k(x_{i,j}) = \max(0, z_k(x_{i,j})) + 1, \quad \alpha_0(x_{i,j}) = \sum_{k=1}^K \alpha_k(x_{i,j})
> $$
>
> We place a Dirichlet prior $\mathrm{Dir}(\boldsymbol{\alpha}(x_{i,j}) + 1)$ over the categorical probabilities $\mathbf{p} \sim \mathrm{Dir}(\boldsymbol{\alpha}(x_{i,j}) + 1)$. AU is defined as the expected predictive entropy:
>
>
> $$
> AU(x_{i,j}) := \mathbb{E} \left[ - \sum_{k=1}^K p_k \log p_k \right]
> $$
>
>
> For a Dirichlet distribution, this expectation has a known closed form, which yields Eq. (4):
>
> $$
> AU(x_{i,j}) = - \sum_{k=1}^K \frac{\alpha_k(x_{i,j})}{\alpha_0(x_{i,j})} \left( \psi(\alpha_k(x_{i,j}) + 1) - \psi(\alpha_0(x_{i,j}) + 1) \right)
> $$
>
> where $\psi(\cdot)$ is the **digamma function**.
> **Thus, the digamma function directly encodes how AU changes with the evidence parameters $\boldsymbol{\alpha}$.**
>
>
> **3.2 Why AU captures “multi-answer answers (ambiguity of tokens)”?**
>
> Let $s := \alpha_0(x_{i,j})$ be the total evidence and $\beta_k := \alpha_k(x_{i,j}) / s$ the evidence shares $\left( \sum_k \beta_k = 1 \right)$. Then AU can be written as $AU(s, \beta)$ in Eq.15.
>
> Two key properties shown in **Appendix A.1** make AU suitable for capturing this token ambiguity:
>
> - **Low evidence cannot fake high AU**: (Proposition A.4). When the total evidence $s$ is small (all logits are weak), AU scales like $O(s)$ and is therefore necessarily small. Thus, high AU cannot arise from a low-confidence, almost–uniform distribution.
>
>   $$
>   AU(s, \beta) = \psi'(1) \cdot s \left(1 - \sum_k \beta_k^2 \right) + O(s^2) \leq \frac{\pi^2}{6} \cdot s + O(s^2)
>   $$
>
> - **Large AU forces evidence dispersion across tokens**: (Proposition A.5). For any fixed $s > 0$ we have
>
>   $$
>   AU(s, \beta) \geq \psi(s + 1) - \psi\left(s \sum_k \beta_k^2 + 1\right)
>   $$
>
>   which implies that if $AU(s, \beta) \geq \eta > 0$, then $\sum_k \beta_k^2$ must be small. Since $\sum_k \beta_k^2 \approx 1 / m$ when the evidence is distributed over $m$ comparable candidates, this lower bound means that a large AU requires a **minimum number of active candidates** carrying non-negligible evidence. In other words, whenever AU is large, the evidence cannot be concentrated on a single token; it must be spread across multiple strong candidates.
>
> Putting these pieces together, AU is large only when both (i) the total evidence $s$ is sufficiently large (**the model is genuinely confident at this position**), and (ii) the evidence shares $\beta$ are well dispersed across several tokens (**multiple strong candidates**). Therefore, $AU(x_{i,j})$ acts as a detector of tokens whose next word admits multiple plausible answers **supported by substantial evidence**, exactly matching our desideratum for ambiguous tokens.
>
>
> **Reference**
>
> [1] Ma, H., Chen, J., Zhou, J. T., Wang, G., & Zhang, C. (2025). Estimating LLM Uncertainty with Evidence. arXiv preprint arXiv:2502.00290.

---

### Official Review · Reviewer_rRdz · 2025-11-05

**Soundness:** 2
**Presentation:** 1
**Contribution:** 2
**Rating:** 2
**Confidence:** 4

**Summary:**

This paper proposes a data selection methods for LLM fine-tuning that considers both sample-level and token-level utility functions. It first scores tokens with two metrics named "Learning Gain" and "Answer Uncertainty," derives sample utilities from these signals, schedules selection with a multi-armed bandit over clusters, and then applies token-aware training (with cross-entropy loss for learnable tokens and a self-distillation loss for ambiguous tokens). The proposed method is evaluated over two base models (Llama3.1-8B and Qwen2-7B) on several benchmarks.

**Strengths:**

- The motivation that data selection should consider both sample-level and token-level utilities makes sense.
- The authors conducted extensive experiments for evaluating the proposed method.
- The empirical results demonstrate a good trade-off between efficiency and efficacy.

**Weaknesses:**

- The method introduces multiple components: two token-level utilities (Learning Gain and Answer Uncertainty), a sample-level utility constructed from token signals, clustering with a bandit scheduler, and token-aware training with self-distillation. However, **the paper does not sufficiently justify why these choices are preferable to alternatives**. Additionally, the large number of components also introduce many hyperparameters. It is unclear how these hyperparameters are selected. And how sensitive is the method with respect to all the hyperparameters.
  + *Token-level utilities.* Please articulate selection principles (e.g., “estimate marginal performance improvement per token under a compute budget”) and show that LG and AU are faithful estimators of those principles. Why these two and not, for example, entropy/MI/gradient-norm/curvature/variance-of-loss across augmentations or checkpoints? A small ablation replacing AU with entropy or temperature-scaled entropy, and replacing LG with gradient-norm or loss delta under adversarial/noisy perturbations, would clarify uniqueness.
  + *Sample-level utility.* The current definition reads ad hoc. Give a semantic interpretation (e.g., “expected marginal improvement per token budget”) and explain why the chosen normalization is appropriate.
  + *Clustering + bandit.* Specify how you embed and cluster data (embedding source, distance metric, k, preprocessing), and why UCB over clusters (vs. e.g., Thompson sampling). Provide a sensitivity analysis on k, UCB parameters, and update cadence.
  + *Self-distillation loss.* Please explain why self-distillation loss is used for the "ambiguous" tokens.
  + *Hyperparameters.* The paper should (i) list all hyperparameters in one table with default values and search ranges; (ii) state the selection protocol (validation objective, budget, and compute); and (iii) report robustness via one-factor-at-a-time curves.
- The mathematical notations are broken here and there. For example, what is $t_{i,j}$ and $l_t$ in Eq (1)? In Def 2, the logits should be written as a function of $t_{i,j}$. The switch from LG to $\Delta l_{i,j}$ should be explicitly defined. What is $\Delta l_{4i, j}$ in Eq (2)? The letter $t$ is overridden for different things.

Overall, the paper currently reads as a collection of plausible components without a sufficiently principled thread tying the choices together. The writing needs significantly improved to convincingly articulate the design choices of the proposed method.

**Questions:**

See Weaknesses.

---

> ### Author Response · Authors · 2025-11-27
> **Response to Reviewer rRdz (1/6)**
>
> > **W1: Overall, the paper currently reads as a collection of plausible components without a sufficiently principled thread tying the choices together. The writing needs to be significantly improved to convincingly articulate the design choices of the proposed method.**
>
> **Response: Clarifying the Design Principles Behind TokenTune**
>
> We thank the reviewer for highlighting the need for a clearer explanation of the guiding design principles behind TokenTune. In the revised manuscript, we have substantially improved the paper, especially **Sections 1 and 2**,  to make the overall logic explicit. In fact, TokenTune is built around three key objectives of data selection, each introducing a distinct challenge. To address these three challenges, we design three corresponding key components, which together form a coherent two-stage pipeline rather than a set of isolated heuristics. We now present the motivation, the unifying design principles, and the way each module follows directly from these principles.
>
> **Part 1: Motivation Behind TokenTune**
>
> We believe that practical data selection methods for instruction tuning should satisfy three fundamental principles:
>
> * **Effectiveness:** maintain or improve the final performance of the model.
> * **Efficiency:** reduce overall training latency, with selection cost offset by fine-tuning savings.
> * **Generalization:**  ensure that the selected subset supports robust performance across diverse downstream benchmarks (tasks).
>
> However, as shown in the table below, many existing approaches fail to meet these criteria. This motivates our **goal**​: to propose a data selection framework that explicitly **meets all three criteria.**
>
> Table I. Selection time and training time across different selection methods on Tulu3 dataset .
> | Method            | Selection Time (h) | Training Time (h) | Total Time (h) |
> |:-----------------:|:------------------:|:------------------:|:--------------:|
> | IFD               |       213          |        3.9         |     216.9      |
> | SelectIT          |       165          |        3.7         |     168.7      |
> | ZIP               |       111          |        3.6         |     114.6      |
> | Entropy           |       129          |        3.4         |     132.4      |
> | Full Data (939k)  |        0           |        152         |     152        |
>
> Table II. Performance across different benchmarks on the Tulu3 dataset
> | Method     | TyDiQA | HellaSwag | ARC-C | BoolQ | GSM8K | Humaneval | LogiQA | Average |
> |:----------:|:------:|:---------:|:-----:|:-----:|:-----:|:---------:|:------:|:-------:|
> | IFD        | 38.55  | 60.35     | 49.87 | 82.21 | 57.04 |   71.60   | 26.36  |  55.14  |
> | SelectIT   | 44.44  | 60.21     | 52.89 | 83.29 | 56.37 |   72.19   | 27.94  |  56.76  |
> | ZIP        | 52.32  | 60.11     | 50.65 | 82.83 | 53.68 |   70.02   | 26.36  |  56.58  |
> | Entropy    | 48.83  | 60.62     | 52.52 | 82.22 | 57.83 |   75.02   | 24.81  |  57.65  |
> | Full Data  | 54.31  | 58.57     | 49.18 | 83.48 | 66.08 |   80.34   | 25.58  | **59.65** |

---

> ### Author Response · Authors · 2025-11-27
> **Response to Reviewer rRdz (2/6)**
>
> **Part 2: Key Challenges and How TokenTune Addresses Them**
>
> To meet the three goals above, we identify three structural challenges in current approaches and explain how TokenTune is intentionally designed to address each one.
>
> **(1) Effectiveness: Limitations of Single-Level Utility Estimation**
>
> Most existing methods perform data selection at either the sample level or the token level. But this ignores a simple truth: a sample can be mostly unhelpful except for a few key tokens, and a useful token may come from a low-scoring sample. Failing to capture the interaction between token-level signals and sample-level context leads to suboptimal selection.
>
> **Key Component 1：Dual-Level Utility Function**
>
> TokenTune addresses this by introducing a dual-level utility function that integrates both granular (token-level) and global (sample-level) perspectives, enabling more precise identification of valuable data.
>
> **(2)Efficiency: Scalability Bottlenecks in Large-Scale Datasets**
>
> While dual-level scoring offers better performance, computing token utility for every sample in a large-scale dataset is prohibitively expensive. This inefficiency limits the scalability of existing methods.
>
> **Key Component 2: Bandit-Guided Cluster Sampling**
>
> TokenTune first embeds and clusters the data pool, then employs a multi-armed bandit (MAB) algorithm to adaptively prioritize and sample promising clusters for scoring. This approach focuses computation on informative samples of the data pool, reducing redundancy and accelerating the selection process without compromising effectiveness.
>
>
> **(3)Generalization: Overfitting Caused by Using Cross-entropy on Multi-answer Tokens**
>
> In standard supervised fine-tuning, cross-entropy loss is applied uniformly to all tokens using one-hot targets. This assumes that each token has a single correct answer. However, instruction tuning often involves multi-answer tokens, where multiple valid responses may exist. Applying hard labels in these cases penalizes plausible alternatives, leading to overfitting and reduced output diversity. A principled selection and training strategy should recognize such multi-answer tokens and incorporate more flexible supervision to preserve generalization.
>
> **Key Component 3: Token-Aware Training for Generalization**
>
> To tackle the overfitting caused by applying cross-entropy loss to multi-answer tokens, TokenTune introduces a token-level utility metric called Answer Uncertainty (AU) to explicitly identify such multi-answer positions. Unlike standard supervised fine-tuning, which uses hard one-hot labels for all tokens, we apply self-distillation loss on tokens with high AU scores. This loss encourages the model to learn from a soft target distribution generated by a stronger teacher, allowing it to maintain uncertainty and assign probability mass to multiple plausible answers. In doing so, TokenTune mitigates the overfitting risk of committing to a single training label and enhances generalization by supporting output diversity.
>
> **Framework Overview.** TokenTune integrates three core components into a coherent two-stage pipeline:
>
> **Stage 1 : Cluster Selection and Sample Utility Estimation**​
>
> TokenTune first embeds and clusters the dataset, then iteratively uses a **Multi-armed Bandit** to select promising clusters for scoring. Within each selected cluster, it computes both token- and sample-level utilities using the **Dual-level Utility Function**​. The bandit strategy is updated based on the estimated utility, allowing the scoring process to progressively focus on high-value regions of the dataset.
>
> **Stage 2: Token-Aware Training.**
>
> Based on token-level utility scores, TokenTune categorizes tokens into three types (i.e., Learnable, Multi-answer, and Uniformative tokens) and applies two tailored optimization objectives. Specifically, it uses cross-entropy loss for Learnable tokens to provide strong and reliable supervision that guides downstream task learning, and self-distillation for multi-answer tokens to preserve output diversity.

---

> ### Author Response · Authors · 2025-11-27
> **Response to Reviewer rRdz (3/6)**
>
> > **W2: The method introduces multiple components: two token-level utilities (Learning Gain and Answer ​Uncertainty), a sample-level utility constructed from token signals, clustering with a bandit scheduler, and token-aware training with self-distillation. However, the paper does not sufficiently justify why these choices are preferable to alternatives.**
>
> **Response: The Uniqueness of TokenTune**
>
> Thank you for this insightful comment. We believe that our **response to W1** has clarified the overall design choices underlying TokenTune. Below, we provide a detailed explanation of your follow-up questions in detail to further justify the motivation and the uniqueness of each components.
>
>
> > **W2.1: Token-level utilities. Please articulate selection principles (e.g., “estimate marginal performance improvement per token under a compute budget”) and show that LG and AU are faithful estimators of those principles. Why these two and not, for example, entropy/MI/gradient-norm/​curvature/variance-of-loss across augmentations or checkpoints? A small ablation replacing AU with entropy or temperature-scaled entropy, and replacing LG with gradient-norm or loss delta under adversarial/noisy perturbations, would clarify uniqueness.**
>
> Based on your suggestion, we conducted an additional ablation study comparing our design with several well-motivated alternatives: (1) replacing LG with Noisy Loss (loss delta under perturbations), and (2) replacing AU with Entropy. As shown in teh table below, TokenTune (LG + AU) consistently achieves the best performance compared to these variations. Below, we will demonstrate the uniqueness of our Token-level Utilities metric through a detailed analysis of the results.
>
>
> **(1) Why LG rather than Gradient Norm or Loss Delta under Adversarial/Noisy Perturbations (LDP)?**
>
> * **Efficiency over Gradient Norm**​: While gradient norm is a reasonable proxy for token informativeness, computing it requires a full backward pass for every sample, which is computationally prohibitive at LLM scale. As noted by LESS [1], fine-grained selection is indeed valuable, but performing such selection through gradient-based computation is infeasible for large models unless substantial approximation or acceleration strategies are introduced. LG provides exactly such an efficient approximation: it is a forward-only, loss-based proxy that captures similar learning dynamics through the delta of token losses, achieving fine-grained estimation without incurring the cost of gradient computation.
> * ​**Effectiveness over Loss Delta under Adversarial/Noisy Perturbations (LDP)**: Although LDP is efficient to compute, it primarily reflects model sensitivity to perturbations and lacks awareness of the model’s current learning state. In contrast, LG captures the model’s evolving uncertainty and identifies tokens with high expected improvement, making it more effective for prioritizing informative tokens during training.
>
>
> **(2) Why AU Instead of Entropy?**
>
> The goal of AU is to identify multi-answer tokens, i.e., positions that admit multiple correct next-token candidates. A distinguishing characteristic of these tokens is that the model assigns high logits to a few distinct candidate tokens, indicating strong confidence in multiple plausible answers. However, standard entropy cannot reliably identify this pattern, because it depends only on the flatness of the softmax distribution. This means that as long as the predicted probabilities over multiple candidate tokens are similar, it will assign a high entropy value, regardless of whether the underlying logits are low or high. In contrast, AU can distinguish between confident multi-answer predictions and uncertain low-confidence ones, since it incorporates both the magnitude and dispersion of the evidence encoded in the logits.
>
> We have included this experiment in **Table 7** and discussion in **Appendix F.2**
>
>
> | Module              | Replace Strategy | TyDiQA | HellaSwag | ARC-C | BoolQ | GSM8K | HumanEval | LogiQA | Average |
> |:-------------------:|:----------------:|:------:|:---------:|:-----:|:-----:|:-----:|:---------:|:------:|:-------:|
> | Token Utility (LG+AU) | LG + AU         | 57.16  | 61.55     | 53.92 | 84.40 | 60.49 |   76.09   | 28.37  | **60.28** |
> |                     | LG + Entropy     | 57.01  | 61.58     | 52.37 | 83.76 | 57.73 |   74.14   | 25.58  | 58.88 |
> |                     | LDP + AU         | 56.72  | 60.17     | 52.02 | 83.79 | 57.16 |   73.94   | 26.19  | 58.57 |
> |                     | LDP + Entropy    | 48.14  | 61.09     | 52.41 | 82.42 | 55.79 |   74.01   | 24.91  | 56.97 |
>
>
> **Reference**
>
> [1] Xia, Mengzhou, Sadhika Malladi, Suchin Gururangan, Sanjeev Arora, and Danqi Chen. "LESS: selecting influential data for targeted instruction tuning." In Proceedings of the 41st International Conference on Machine Learning.

---

> ### Author Response · Authors · 2025-11-27
> **Response to Reviewer rRdz (4/6)**
>
> > **W2.2:Sample-level utility. The current definition readsad hoc. Give a semantic interpretation (e.g., “expected marginal improvement per token budget”) and explain why the chosen normalization is appropriate.**
>
> **Response**
>
> We thank the reviewer for pointing this out. In the revised version (see​ **Section 2**​), we have clarified the semantics of our sample-level utility function and provided a principled explanation for the chosen normalization strategy.
>
> **[Semantic Interpretation]** Our sample-level utility is designed to estimate the expected marginal benefit of each example under a constrained token budget. Semantically, it captures the average informativeness per token. In other words, it refers to how much learning signal a sample provides relative to its length.
>
> **[Explanation of the chosen normalization​]** To eliminate length-induced bias, we normalize the total token utility by the number of tokens in each sample. This ensures that selection prioritizes samples that are dense in learning signal, rather than those that are simply longer. To suport our justification, we compare the asequence length of the Tulu3 dataset ranked by either un-normalized and normalized score functions. It is apparent that without normalization, data with the highest scores(low ranking) correspond to data with very short sequence lengths. In contrast, the introduction of normalization completely removes this phenomenent.
>
> **[Additional Experiment]**  We conducted a comparative experiment with and without normalization. As shown in the table below, removing normalization leads to a consistent performance drop (average: 60.07 → 58.24), highlighting the importance of this adjustment in effective sample selection. We have included this experiment in **Table 8** and discussion in **Appendix F.3**
>
>
> | Module        | Replace Strategy     | TyDiQA | HellaSwag | ARC-C | BoolQ | GSM8K | HumanEval | LogiQA | Average |
> |:-------------:|:--------------------:|:------:|:---------:|:-----:|:-----:|:-----:|:---------:|:------:|:-------:|
> | Sample Utility | Normalization        | 57.16  | 61.55     | 53.92 | 84.40 | 60.49 |   76.09   | 28.37  | **60.28** |
> |               | w/o Normalization    | 57.15  | 61.47     | 51.97 | 83.66 | 55.44 |   72.43   | 25.58  | 58.24 |
>
>
>
> > **W2.3: Specify how you embed and cluster data (embedding source, distance metric, k, preprocessing), and why UCB over clusters (vs. e.g., Thompson sampling).**
>
> **Response**
>
> Thank you for the helpful question. We have clarified these implementation details in **Appendix C.3** of the revised paper.
>
> Specifically, we use the pre-trained encoder BAAI/bge-en-v1.5 to generate embeddings for all training samples, and apply K-Means clustering with cosine similarity as the distance metric, setting k = 1000.
>
> As for the use of UCB for cluster selection, we would like to clarify that our choice is not based on a preference for a specific bandit algorithm. Instead, our goal is to adopt the general idea of exploration–exploitation from the MAB framework to prioritize the sample utility estimation process on promising clusters, thereby reducing redundant computation on low-quality clusters and accelerating selection.
>
> To support this claim, we have conducted an ablation study comparing UCB, Thompson Sampling, and EXP3 within our MAB-based selection pipeline. As shown in the table below, all three methods yield similar performance, suggesting that the effectiveness of our approach stems from the use of MAB-based sampling itself, rather than any specific bandit algorithm choice. We have included this experiment in **Table 9** and discussion in **Appendix F.4**
>
>
> | Module  | Replace Strategy | TyDiQA | HellaSwag | ARC-C | BoolQ | GSM8K | HumanEval | LogiQA | Average |
> |:-------:|:----------------:|:------:|:---------:|:-----:|:-----:|:-----:|:---------:|:------:|:-------:|
> | **MAB** | UCB              | 57.16  | 61.55     | 53.92 | 84.40 | 60.49 |   76.09   | 28.37  | **60.28** |
> |         | Thompson         | 57.21  | 61.47     | 53.86 | 83.98 | 58.97 |   77.12   | 27.41  | 60.00 |
> |         | EXP3             | 58.19  | 61.52     | 53.92 | 83.97 | 59.18 |   76.09   | 27.62  | 60.07 |
> | **Cluster** | Kmeans       | 57.16  | 61.55     | 53.92 | 84.40 | 60.49 |   76.09   | 28.37  | **60.28** |
> |         | DBSCAN           | 57.93  | 61.07     | 54.17 | 83.26 | 57.82 |   76.48   | 27.95  | 59.81 |
> |         | GMM              | 56.14  | 61.53     | 54.89 | 84.23 | 59.25 |   76.92   | 28.30  | 60.18 |

---

> ### Author Response · Authors · 2025-11-27
> **Response to Reviewer rRdz (5/6)**
>
> > **W2.4: Please explain why self-distillation loss is used for the "ambiguous" tokens**
>
> **Response**
>
> Thank you for this insightful comment. We acknowledge that our initial submission did not sufficiently explain this design choice, and we have revised **Section 2.4** in the updated version to clarify our motivation. Below, we provide a detailed explanation, including the intuition, a concrete example, and supporting ablation results.
>
> **[Detailed Explaination]** In our setting, ambiguous tokens refer to positions where the model is confident that multiple distinct tokens could be correct (i.e., multi-answer tokens). For these tokens, using standard cross-entropy loss introduces a key limitation, because it uses a ​**one-hot target distribution**​, where the observed token is assigned a probability of 1, and all other tokens are treated as incorrect with probability 0. To minimize this loss, the model is trained to assign maximal probability to the observed token while reducing probability for all alternatives. Over time, this causes the model to overfit to the single training label, even when other valid answers exist, suppressing semantic diversity and limiting generalization.
>
> To address this, we apply self-distillation loss at ambiguous positions. Instead of optimizing toward a single hard label, self-distillation aligns the student model’s output with the **soft target distribution** generated by a stronger teacher. This distribution naturally assigns non-zero probability to multiple plausible candidates, enabling the student to retain uncertainty and share probability mass across alternatives.
>
> **[Example]** To illustrate this behavior, we provide an example.
>
> ​**Question**​: "What's a good drink to start the day?"
>
> **Answer in ​Training Data**​: "Coffee"
>
> In this case, the model may assign high confidence to both “coffee” and “tea”​, as both are reasonable answers. However, since the training data only contains “coffee”​, cross-entropy would penalize “tea”​, even though it is equally valid. In contrast, self-distillation allows the model to preserve probability mass across both, maintaining flexibility and generalization under multi-answer supervision.
>
> **[Additional Experiment]** To validate the effectiveness of self-distillation on ambiguous tokens, we conduct an ablation by replacing the distillation loss with standard cross-entropy loss on all tokens. As shown below, removing self-distillation (i.e., applying CE on all tokens) leads to a consistent drop in performance across benchmarks, with the average score decreasing from 60.28 to 58.72. This confirms the importance of using soft supervision on ambiguous tokens to preserve answer diversity and improve generalization. We have included this experiment in **Table 10** and discussion in **Appendix F.5**
>
>
> | Module   | Replace Strategy      | TyDiQA | HellaSwag | ARC-C | BoolQ | GSM8K | HumanEval | LogiQA | Average |
> |:--------:|:---------------------:|:------:|:---------:|:-----:|:-----:|:-----:|:---------:|:------:|:-------:|
> | SD Loss  | CE (LG) + SD (AU)     | 57.16  | 61.55     | 53.92 | 84.40 | 60.49 |   76.09   | 28.37  | 60.28   |
> |          | CE (LG+AU)            | 57.13  | 61.32     | 52.19 | 82.47 | 55.72 |   75.42   | 26.81  | 58.72   |
>
>
> > **W4: - The mathematical notations are broken here and there.  For example, what is $t_{i,j}$ and $l_t$ in Eq (1)?  In Def 2, the logits should be written as a function of $t_{i,j}$.  The switch from LG to $\Delta l_{i,j}$ should be explicitly defined.  What is $\Delta l_{i,j}$ in Eq (2)?  The letter $t$ is overridden for different things.**
>
> **Response**
>
> We appreciate the reviewer’s feedback, which helped improve the clarity of our presentation. In the revised version, we have carefully reviewed and standardized all mathematical notations, ensuring consistency across equations and definitions. Please refer to **Section 2** and **Section 3** of the updated paper for more details. We additionally provide a complete and unified notation table in **Appendix K** to further enhance readability and ensure unambiguous interpretation of all symbols used throughout the paper.

---

> ### Author Response · Authors · 2025-11-27
> **Response to Reviewer rRdz (6/6)**
>
> > **W3: Hyperparameters. The paper should**
> >
> > * **(i) list all hyperparameters in one table with default values and search ranges;**
> > * **​(ii) state the selection protocol (​validation objective, budget, and compute);**
> > * **(iii) report robustness via one-factor-at-a-time curves.**
> >
> > **W4: Provide a sensitivity analysis on k, UCB parameters, and update cadence.**
>
> **Response**
>
> We thank the reviewer for the thoughtful suggestions. We have revised the paper accordingly and added the requested details in **Appendix C** and **Appendix I**​.
>
> ​For (i), we now provide a complete hyperparameter table (default values + search ranges) in **Appendix C.4**​.
>
> ​For (ii), the full selection protocol, including validation objective, budget, compute setup, clustering, and MAB scheduling, has been clarified in **Appendix C.3**​.
>
> ​For (iii)​, we provide one-factor-at-a-time robustness curves for all major hyperparameters in ​**Appendix I**​, together with detailed explanations.
>
> Below, we report sensitivity results for key parameters in TokenTune. The results show that performance is generally stable within local neighborhoods of each hyperparameter, confirming the robustness of our method. Please refer to **Appendix I** for more details.
>
>
> Table I. Effect of Thresholds for Token Utility.
> | Threshold (Token Utility) | TyDiQA | HellaSwag | ARC-C | BoolQ | GSM8K | HumanEval | LogiQA | Average |
> |:-------------------------:|:------:|:---------:|:-----:|:-----:|:-----:|:---------:|:------:|:-------:|
> | 0.7                       | 57.78  | 61.56     | 52.54 | 84.00 | 58.04 |   74.54   | 26.82  | 59.33   |
> | 0.6                       | 57.16  | 61.55     | 53.92 | 84.40 | 60.49 |   76.09   | 28.37  | **60.28** |
> | 0.5                       | 57.50  | 61.69     | 53.57 | 84.03 | 57.81 |   75.93   | 27.13  | 59.67   |
> | 0.4                       | 55.88  | 61.45     | 52.89 | 84.37 | 57.50 |   73.52   | 24.65  | 58.61   |
> | 0.3                       | 54.17  | 60.02     | 51.38 | 82.47 | 55.73 |   74.62   | 25.78  | 57.74   |
>
>
> Table II. Effect of Sample Budgets.
> | Sample Budgets | TyDiQA | HellaSwag | ARC-C | BoolQ | GSM8K | HumanEval | LogiQA | Average |
> |:--------------:|:------:|:---------:|:-----:|:-----:|:-----:|:---------:|:------:|:-------:|
> | 1%             | 51.66  | 61.56     | 54.61 | 84.43 | 54.13 |   74.72   | 27.29  | 58.34   |
> | 2.5%           | 55.78  | 61.03     | 53.74 | 83.94 | 56.28 |   75.13   | 28.01  | 59.13   |
> | 5%             | 57.16  | 61.55     | 53.92 | 84.40 | 60.49 |   76.09   | 28.37  | **60.28** |
> | 7%             | 54.10  | 61.64     | 54.44 | 85.89 | 57.89 |   74.36   | 27.60  | 59.42   |
> | 10%            | 59.15  | 61.13     | 53.40 | 83.89 | 57.20 |   76.04   | 26.82  | 59.66   |
>
>
>
> Table III. Effect of Different Clustering Algorithm.
> | Different Clustering Algorithm | GMM | DBSCAN | KMeans   |
> |:-----------------------------:|:------:|:------:|:-----:|
> | Avg. Performance              | 58.73  | 58.91  | **60.28** |
>
>
>
> Table IV. Effect of MAB Arms / Clustering Size.
> | MAB Arms / Clustering Size K | 100   | 500   | 1000  | 2000  | 3000  | 5000  |
> |:----------------------------:|:-----:|:-----:|:-----:|:-----:|:-----:|:-----:|
> | Performance (Avg.)          | 58.73 | 58.91 | **60.28** | 60.01 | 59.67 | 59.04 |
>
>
> | Exploration Rate γ of MAB | 0.0005 | 0.001  | 0.005  | 0.01   | 0.05   |
> |:--------------------------:|:------:|:------:|:------:|:------:|:------:|
> | Avg. Performance           | 59.02  | **60.28** | 59.67  | 59.04  | 57.88  |
>
>
> Table V. Effect of Update Cadence.
> | Update Cadence (Sample Ratio / Iteration) | 0.5%  | 1%     | 2.5%   | 5%     |
> |:-----------------------------------------:|:-----:|:------:|:------:|:------:|
> | Avg. Performance                          | 60.12 | **60.28** | 59.87  | 59.45  |

---

### Author Response · Authors · 2025-11-27
**General Response**

We sincerely thank all reviewers for their great efforts and insightful comments, based on which we have significantly improved the paper and clarified the concerns from the following two parts. The revised parts in the paper are highlighted in **blue**.

**Part 1: New Experiments.** We have added **15 additional experiments** ​(e.g., Parameter Sensitivity Analysis, Generalization analysis and Ablation Study of Design Space) to demonstrate the effectiveness, efficiency and generalization of TokenTune.

**Part 2: New Changes.** We have also made the following main changes in the revised manuscript:

* (C1) We rewrote **Section 1** and **Section 2.1** to make the overall logic explicit and clearly connect the three goals of data selection with the structural challenges and the corresponding three components of TokenTune.

* (C2) We revised **Sections 2.2** and **2.3**​, and added explanations in **Appendix J**​, to clarify the selection principles and justify the uniqueness of TokenTune's design choice.
* (C3) We uploaded an improved **Figure 2** to more clearly illustrate the full two-stage pipeline of TokenTune.
* (C4) We added new tables: ​**Table 3, 5, 6, 7, 8, 9, 10, 11, 12, 13, 14, 15, 16, 17**​, covering design-space ablations, budget scaling, generalization results, sensitivity studies, and compute breakdowns.
* (C5) We enhanced the justification of TokenTune’s **effectiveness** by adding comprehensive analyses under multiple budget levels, various dataset, different model size and different architectural settings in **Appendix D**, **Appendix G** and ​**Appendix H**​.
* (C6) We added **new generalization analyses** in ​**Appendix H**​, including cross-architecture ( Llama-3.1-8B → Qwen2-7B) and cross-scale (GPT2 → Llama-3.1-8B) transfer settings.
* (C7) We included extensive sensitivity analyses in **Appendix I** for token utility thresholds, clustering size, budget ratio, exploration rate, and update cadence.
* (C8) We added complete implementation details and hyperparameter tables (default values + search ranges) in **Appendix C** and **Appendix J** to ensure full reproducibility.
* (C9) We carefully revised and standardized all mathematical notations and provided a full notation table in **Appendix K**​.

---

### Author Response · Authors · 2025-12-03
**Rebuttal Summary (3/3)**

We believe these revisions fully address the reviewers’ concerns and significantly strengthen the submission. Unfortunately, due to the substantial volume of new experiments and limited computational resources, we were only able to submit our full rebuttal and updates by November 27, just before the unexpected early closure of the discussion phase, which left us without the opportunity to further engage with the reviewers.


We fully understand that the system rollback has created additional burden for the new Area Chair. We provide this brief summary only in the hope of presenting as clearly as possible the current state of the paper and the efforts we have made to address the reviewers’ comments.  Regardless of the final decision, we sincerely thank you for your time and for reviewing our work amid your busy schedule.


Best regards,

The Authors

---

### Author Response · Authors · 2025-12-03
**Rebuttal Summary (2/3)**

**Part 2: Additional Experiment**

The main concerns raised by the three reviewers can be summarized into the following five points:

- (C1) Lack of generalization evidence across model sizes, architectures, and large-scale datasets
- (C2) Lack of justification of the uniqueness for design choices
- (C3) Lack of validation on robustness to hyperparameters
- (C4) Scalability concerns under larger candidate pools and varying sample budgets
- (C5) Insufficient clarity on rationale for individual modules

We have added **15 new experiments** to address these concerns. These experiments are summarized below and detailed in the revised manuscript in **Appendix D–I**, specifically in **Tables 3, 5, 6, 7, 8, 9, 10, 11, 12, 13, 14, 15, 16, 17**.

| **Type**                                  | **Experiment**                             | **Purpose**                                                                 | **Addressed Concerns** | **Requests**              |
|-------------------------------------------|---------------------------------------------|------------------------------------------------------------------------------|--------------------------|---------------------------|
| **D. Overall Performance on OpenHermes**   | D.1 Effectiveness on OpenHermes Dataset     | Evaluate TokenTune’s performance on a large-scale dataset                   | (C1), (C4)               | L7K1 [Q1]                 |
| **E. Additional Backbone on Tulu3 Dataset**| E.1 Performance on LLaMA-3B                 | Evaluate TokenTune on a smaller backbone                                    | (C1)                     | BxaP [W2]                 |
|                                           | E.2 Performance on LLaMA-13B                | Evaluate TokenTune on a larger backbone                                     | (C1)                     | BxaP [W2]                 |
| **F. Detailed Ablation Study of Design Space** | F.2 Token-level Utility Function        | Compare LG with alternative token-utility metrics                           | (C2)                     | rRdz [W2.1]               |
|                                           | F.3 Sample-level Utility Function           | Evaluate the effect of normalization and formulation choices                | (C2), (C5)               | rRdz [W2.2]               |
|                                           | F.4 MAB Module                              | Evaluate the impact of different MAB algorithms on sampling effectiveness   | (C2), (C3)               | rRdz [W2.3]               |
|                                           | F.5 Self-Distillation Loss                  | Assess the contribution of self-distillation loss on multi-answer tokens    | (C5)                     | rRdz [W2.4]               |
| **G. Detailed Analysis of Data Scaling**   | G.1 Sample Size Scaling                     | Evaluate generalization across varying fine-tuning sample budgets           | (C1), (C4)               | L7K1 [W2]                 |
|                                           | G.2 Pool Size Scaling                       | Examine TokenTune’s performance as the candidate pool size increases        | (C4)                     | L7K1 [W2]                 |
| **H. Generalization of TokenTune**         | H.1 Cross-Architecture Setting              | Evaluate generalization across different architectures                      | (C1)                     | L7K1 [W2]                 |
|                                           | H.2 Cross-Scale Setting                     | Evaluate generalization across different model sizes                        | (C1)                     | BxaP [W2]                 |
| **I. Parameter Sensitivity Analysis**      | I.1 Token Utility Threshold                 | Assess robustness to token-threshold choices                                | (C3)                     | rRdz [W3], L7K1 [W2]      |
|                                           | I.2 MAB Exploration Rate                    | Analyze exploration-exploitation balance                                   | (C3)                     | rRdz [W2.3], L7K1 [W2]    |
|                                           | I.3 Number of MAB Clusters (k)              | Examine sensitivity to number of clusters (MAB arms)                        | (C3)                     | rRdz [W2.3], L7K1 [W2]    |
|                                           | I.4 Clustering Algorithm                    | Evaluate sensitivity across clustering algorithms                           | (C3)                     | rRdz [W2.3], L7K1 [W2]    |

---

### Author Response · Authors · 2025-12-04
**Rebuttal Summary (1/3)**

Dear Area Chair,

Thank you for reviewing our submission TokenTune. During the discussion phase, we provided detailed responses to all reviewer concerns, conducted 15 additional experiments, and revised the manuscript extensively to improve clarity, completeness, and empirical validation.

This summary provides a comprehensive overview of the current status of the paper, including:

(1) **a structured breakdown of reviewer concerns and our clarifications or improvements in response**;

(2) **a summary of new experiments and analyses** that directly address core issues such as scalability, design justification, robustness, and generalization.

All new experiments are presented in the updated manuscript (Appendix D–I), with results shown in Tables 3 and 5–15.

---

**Part 1: Summary of Reviewers' Concerns and Our Responses**

To clarify how we addressed the key concerns raised during the review process, we summarize below the main concerns from each reviewer and our corresponding responses.


**Reviewer rRdz:** Main concerns focused on presentation, raising concerns that the connections among components, hyperparameters, and notation could be articulated more clearly.  We addressed these directly by:

- Rewrote **Sections 1 and Sections 2.1** to clarify the motivation and design logic of TokenTune, aligning three core objectives with corresponding challenges and modules, and presenting the method as a principled pipeline rather than a collection of heuristics.
- Revised **Sections 2.2** and **2.3**, and added detailed explanations in **Appendix J**, to clarify the selection principles and justify the uniqueness of TokenTune's design choice.
- Uploaded an improved Figure 2 to more clearly illustrate the full two-stage pipeline of TokenTune.
- Added ablation studies of design space for each module to justify the design of TokenTune (**Appendix F**)
- Standardized all notations and provided a complete notation table (**Appendix K**)


**Reviewer BxaP:** Main concerns focused on experimental completeness and methodological clarity, raising concerns about missing error bars, unspecified MAB hyperparameters, the rationale behind AU for multi-answer tokens, pruning effects, and evidence for scalability across model sizes.  We addressed these directly by:
- Added new experiments using both smaller (3B) and larger (13B) backbones to demonstrate robustness across model sizes (**Appendix H**; **Table 8**)
- Clarified that TokenTune performs gradient masking, not token pruning, during backpropagation, which preserves fluency and sentence structure
- Added **theoretical analysis** on the AU metric and its link to entropy and variance, explaining the motivation behind using the digamma function (**Section 2.2, Section 3.1, Appendix A**)
- Provided comprehensive implementation details and sensitivity analyses for the MAB scheduler, including exploration rate, cluster size, and update cadence (**Appendix C, I**; **Tables 6, 13–15**)


**Reviewer L7K1:** Main concerns focused on potential selection bias and experimental completeness, raising concerns about insufficient hyperparameter justification, scalability to larger pools, limited analysis of subset diversity and token-level scoring bias. We addressed these directly by:

- Demonstrated scalability on a 1M+ example dataset (OpenHermes) and across multiple pool sizes (**Appendix D**)
- Provided a **complete compute breakdown**, including token-level scoring, clustering cost, and fine-tuning cost relative to full-data training
- Added extensive sample-scaling experiments (1%, 2.5%, 5%, 7%, 10%), showing consistent gains across all budget levels (**Appendix G**)
- Clarified that TokenTune ensures diversity at both the sample level (via MAB-guided cluster sampling) and the token level (via AU-based identification of multi-answer tokens)
- Added cross-architecture and cross-scale robustness studies demonstrating that utility scores transfer well even when selection and fine-tuning models differ (**Appendix H**)
- Conducted comprehensive sensitivity studies on token utility thresholds, clustering size, budget ratio, exploration rate, and update cadence (**Appendix I**)

---

### Meta-Review · Area_Chair_mGDi · 2026-01-07

**Summary:**

The paper introduces a data selection framework for LLM instruction tuning that employs a dual-level utility function to evaluate both tokens and samples. It integrates a Multi-Armed Bandit scheduler for scalability and a gated optimization strategy that identifies "learnable" and "multi-answer" tokens. While the proposed framework demonstrates notable empirical effectiveness across several benchmarks, the consensus among reviewers is that the methodology represents an integration of multiple heuristic components whose synergy lacks the unified conceptual foundation expected for a fundamental algorithmic advancement.

**Reviewer Concerns:**

Addressed:
- The authors demonstrated an extraordinary level of effort during the rebuttal phase, conducting additional experiments to clarify technical aspects of the pipeline.
- These included scalability tests on the 1M-sample OpenHermes dataset, generalization studies across model sizes, and extensive sensitivity analyses.
- The authors also standardized mathematical notations and provided a comprehensive notation table in the appendix.


Outstanding:
- While the authors attempted to re-align the framework's logic around three core objectives in the revised Sections, the synergy between MAB scheduling, dual-level scoring, and self-distillation is still perceived as a heuristic integration of solutions for separate issues rather than a unified algorithmic advancement.
- Reviewers maintain that the design choices appear more as an effective assembly of empirical optimizations than a cohesive scientific narrative.
- Despite the addition of theoretical support for the Answer Uncertainty metric, skepticism persists regarding whether the complexity of the proposed indicators provides fundamental scientific insights into data utility beyond what is achievable with simpler, established baselines.

**Reviewer Scores:**

Reviewer rRdz: Maintained a negative recommendation, emphasizing that the work functions primarily as an engineering optimization lacking a cohesive scientific thread.

Reviewer BxaP: Acknowledged the creativity of the work but remained unconvinced by the necessity and clarity of the methodological rationale.

Reviewer L7K1: Leaned positive based on the empirical results and the authors' responsiveness, though noted lingering conceptual questions regarding selection bias and diversity.

---

### Decision · Program_Chairs · 2026-01-26

Reject